# ADAM10 and ADAM17 promote SARS-CoV-2 cell entry and spike protein-mediated lung cell fusion

Georg Jocher[1,2,†] ⓘ, Vincent Grass[3,†] ⓘ, Sarah K Tschirner[1,2], Lydia Riepler[4] ⓘ, Stephan Breimann[1,2,5] ⓘ, Tuğberk Kaya[1,6,7], Madlen Oelsner[8], M Sabri Hamad[3] ⓘ, Laura I Hofmann[1,2], Carl P Blobel[9,10] ⓘ, Carsten B Schmidt-Weber[8] ⓘ, Ozgun Gokce[7,11], Constanze A Jakwerth[8], Jakob Trimpert[12] ⓘ, Janine Kimpel[4] ⓘ, Andreas Pichlmair[3,13,‡] ⓘ & Stefan F Lichtenthaler[1,2,11,*,‡] ⓘ

## Abstract

The severe-acute-respiratory-syndrome-coronavirus-2 (SARS-CoV-2) is the causative agent of COVID-19, but host cell factors contributing to COVID-19 pathogenesis remain only partly understood. We identify the host metalloprotease ADAM17 as a facilitator of SARS-CoV-2 cell entry and the metalloprotease ADAM10 as a host factor required for lung cell syncytia formation, a hallmark of COVID-19 pathology. ADAM10 and ADAM17, which are broadly expressed in the human lung, cleave the SARS-CoV-2 spike protein (S) *in vitro*, indicating that ADAM10 and ADAM17 contribute to the priming of S, an essential step for viral entry and cell fusion. ADAM protease-targeted inhibitors severely impair lung cell infection by the SARS-CoV-2 variants of concern alpha, beta, delta, and omicron and also reduce SARS-CoV-2 infection of primary human lung cells in a TMPRSS2 protease-independent manner. Our study establishes ADAM10 and ADAM17 as host cell factors for viral entry and syncytia formation and defines both proteases as potential targets for antiviral drug development.

**Keywords** A549; apratastat; DPC-333; ectodomain shedding; syncytia formation

**Subject Categories** Microbiology, Virology & Host Pathogen Interaction; Molecular Biology of Disease; Post-translational Modifications & Proteolysis

## Introduction

The coronavirus-induced disease 2019 (COVID-19) pandemic is caused by the severe acute respiratory syndrome coronavirus 2 (SARS-CoV-2), a membrane-enveloped RNA virus. Through its surface spike (S) protein, SARS-CoV-2 binds to surface receptors on host cells, mainly the angiotensin-converting enzyme 2 (ACE2; Hoffmann *et al*, 2020b; Zhou *et al*, 2020). S is a furin protease-cleaved heterodimer consisting of an S1 subunit that contains the receptor-binding domain and an S2 subunit that is proteolytically primed at an S2' site for fusion with the host cell. Protease-mediated S2 priming by transmembrane protease serine 2 (TMPRSS2) or furin or other, as yet unknown, host cell proteases is assumed to enhance the fusion of the viral SARS-CoV-2 particle with the plasma membrane of the host cell (preprint: Essalmani *et al*, 2020; Hoffmann *et al*, 2020a; Matsuyama *et al*, 2020; Shang *et al*, 2020). In an alternative cell entry pathway that requires viral endocytosis and low pH, S2 priming may be mediated by cathepsins to allow fusion with the endo-lysosomal membrane (Hoffmann *et al*, 2020a; Ou *et al*, 2020; Shang *et al*, 2020).

Proteolysis does not only occur for proteins on the viral surface, but also for proteins on the host cell surface, where membrane-bound proteases, such as 'a disintegrin and metalloproteases' (ADAMs), beta-site APP cleaving enzymes (BACEs), and TMPRSS2 can proteolytically shed the ectodomain of surface proteins. This is a basic cellular mechanism controlling the abundance and activity of membrane proteins, including hundreds of surface

---

1  German Center for Neurodegenerative Diseases (DZNE), Munich, Germany
2  Neuroproteomics, School of Medicine, Klinikum rechts der Isar, Technical University of Munich, Munich, Germany
3  School of Medicine, Institute of Virology, Technical University of Munich, Munich, Germany
4  Department of Hygiene, Microbiology and Public Health, Institute of Virology, Medical University of Innsbruck, Innsbruck, Austria
5  Department of Bioinformatics, Wissenschaftszentrum Weihenstephan, Technical University of Munich, Freising, Germany
6  Institute of Neuronal Cell Biology, Technical University Munich, Munich, Germany
7  Institute for Stroke and Dementia Research (ISD), University Hospital, LMU Munich, Munich, Germany
8  Center of Allergy and Environment (ZAUM), Technical University of Munich and Helmholtz Center Munich, German Research Center for Environmental Health, German Center for Lung Research (DZL), Munich, Germany
9  Arthritis and Tissue Degeneration Program, Hospital for Special Surgery, New York, NY, USA
10 Departments of Medicine and of Physiology, Biophysics and Systems Biology, Weill Cornell Medicine, New York, NY, USA
11 Munich Cluster for Systems Neurology (SyNergy), Munich, Germany
12 Institut für Virologie, Freie Universität Berlin, Berlin, Germany
13 German Center for Infection Research (DZIF), Munich partner site, Munich, Germany
   *Corresponding author. Tel: +49 89 4400 46426; E-mail: stefan.lichtenthaler@dzne.de
   †These authors contributed equally to this work
   ‡These authors contributed equally to this work as senior authors

proteins (Lichtenthaler *et al*, 2018). The shed substrates also comprise virus receptors, such as the Coxsackievirus and Adenovirus Receptor (CAR; Houri *et al*, 2013; Kuhn *et al*, 2016), the enterovirus 71 receptor P-selectin glycoprotein ligand 1 (PSGL-1; Lichtenthaler *et al*, 2003; Nishimura *et al*, 2009), the SARS-CoV-2 coreceptor NRP1 (Romi *et al*, 2014) and ACE2, the main receptor for SARS-CoV-2 and for the related SARS-CoV (Lambert *et al*, 2005; Haga *et al*, 2008; Heurich *et al*, 2014), which caused an epidemic in 2003. In many cases, it remains unclear whether shedding of the viral receptor is functionally relevant for viral entry or merely contributes to receptor degradation, but the shedding of ACE2 by TMPRSS2 can augment viral entry of SARS-CoV (Glowacka *et al*, 2010, 2011; Heurich *et al*, 2014). It is not known whether other host cell shedding proteases can also contribute to infectivity and may be targeted for therapeutic inhibition of SARS-CoV-2 infection.

Shedding of host proteins or SARS-CoV-2 S may also contribute to one of the pathological hallmarks in the lungs of COVID-19 patients, i.e., the abnormal morphology of pneumocytes with frequent multinucleation, indicative of syncytia formation, which is assumed to occur through the fusion of cells expressing the S protein with cells expressing the S receptor ACE2 (Buchrieser *et al*, 2020; Bussani *et al*, 2020; Braga *et al*, 2021). Exogenously added proteases, such as trypsin or overexpressed TMRPSS2, and a yet unknown metalloprotease (Nguyen *et al*, 2020; Hörnich *et al*, 2021), can enhance syncytia formation *in vitro*, but it is unclear whether endogenous shedding proteases are involved and may be suitable as drug targets for pharmacological inhibition of pathological syncytia formation.

Here, we report that the shedding proteases ADAM10 and ADAM17—independently of the protease TMPRSS2—facilitate two S protein-dependent molecular events that contribute to COVID-19 pathogenesis, SARS-CoV-2 infectivity, and SARS-CoV-2-mediated lung cell fusion. We found that infectivity of SARS-CoV-2 is enhanced by ADAM17 through control of the viral entry step into lung cells. Conversely, lung cell syncytia formation was found to require ADAM10. Mechanistically, both ADAM10 and ADAM17 cleaved the SARS-CoV-2 spike protein *in vitro*. Thus, our study establishes ADAM17 and ADAM10 as host cell factors for viral entry and pathological lung cell fusion.

# Results

## Metalloprotease inhibitors reduce SARS-CoV-2 infection of human A549 lung cells

To determine whether membrane protein ectodomain shedding contributes to SARS-CoV-2 infection, we infected the human lung alveolar cell line A549—stably expressing the SARS-CoV-2 receptor ACE2 (Fig 1A)—with a green fluorescent protein (GFP)-expressing version of SARS-CoV-2 (Thi Nhu Thao *et al*, 2020) and evaluated virus growth upon pharmacological inhibition of shedding proteases, using time-resolved live-cell imaging. In agreement with previous studies (Hoffmann *et al*, 2020b; Zhou *et al*, 2020), ACE2 expression allowed SARS-CoV-2 infection, as seen by GFP expression in A549-ACE2, but not wild-type A549 cells (Fig 1B and C). Treatment of cells with the TMPRSS2 inhibitor camostat did not alter GFP expression and served as a negative control because

A549 cells do not express TMPRSS2 (Matsuyama *et al*, 2020; Fig 1D). Conversely, the cathepsin inhibitor E64d, which blocks SARS-CoV-2 fusion after endocytosis, strongly reduced GFP expression and served as a positive control, demonstrating that SARS-CoV-2 can enter human lung cells through an endocytic-lysosomal pathway in agreement with previous reports (Hoffmann *et al*, 2020b; Shema Mugisha *et al*, 2020; Zhu *et al*, 2021). Inhibition of the common shedding proteases BACE1 and BACE2, which are expressed in A549 cells (Zecha *et al*, 2020), with the drug C3 (Stachel *et al*, 2004) did not alter GFP expression, ruling out an involvement of both proteases in SARS-CoV-2 spread in this infection model (Fig 1D). In contrast, the broad-spectrum metalloprotease inhibitors BB94 (batimastat) and TAPI-1, which block the activity of several ADAMs and matrix metalloproteases (MMPs), reduced GFP expression in a dose-dependent manner compared with DMSO treatment and to a similar degree as the cathepsin inhibitor E64d (Fig 1D). Conversely, the phorbol ester PMA, which activates the shedding of numerous metalloprotease substrates, enhanced viral infection as seen by increased GFP expression (Fig 1E). Together, these results indicate a direct or indirect involvement of one or several endogenously expressed ADAMs or MMPs in SARS-CoV-2 infection. To further substantiate this conclusion, we tested two additional ADAM protease-targeting inhibitors, apratastat (TMI-005) and DPC-333 (BMS-561392), which efficiently reduce the ADAM17-mediated shedding of tumor necrosis factor α (TNFα) in humans (Qian *et al*, 2007; Shu *et al*, 2011) and were tested in phase 2 clinical trials for rheumatoid arthritis (Moss *et al*, 2008). In line with the clinical trial data, both drugs potently blocked endogenous TNFα shedding from human U937 macrophages stimulated with lipopolysaccharide (LPS, Fig EV1A). Similar results were obtained for BB94. Importantly, both apratastat and DPC-333 potently attenuated SARS-CoV-2 infection of A549-ACE2 cells, as seen by reduced GFP expression, similarly to BB94 (Fig 1F). A dose-response curve demonstrated that viral inhibition was seen at very low concentrations of 10 nM and higher (Fig 1G). In the micromolar concentration range, drug treatment led to a maximum viral inhibition of more than 50%, indicating that metalloproteases are critical host factors for virus infection but that viral entry is still possible—although to a reduced extent—when metalloproteases are blocked. Potentially other proteases contribute as well or mechanisms independent of proteolytic cleavage. While performing time-resolved live-cell imaging, we observed that pharmacological treatment of A549-ACE2 cells did not affect the growth rate of cells compared with the DMSO control (Fig EV1B and C), indicating that the tested drugs did not alter cell viability at the applied concentrations. This was further confirmed by an MTT viability assay, where the metalloprotease inhibitors did not show significant toxicity at the tested concentrations (Fig EV1D). All inhibitors were used in a concentration range that is known to block ADAM10, ADAM17, and BACE1, as seen by inhibition of sAPPα and sAPPβ (Fig EV1E and F). Both sAPPα and sAPPβ are fragments of the Alzheimer's disease-linked amyloid precursor protein (APP; Fig EV1E and F) that are generated by the proteolytic activity of ADAM10 or ADAM17 (sAPPα) or by BACE1 (sAPPβ; Buxbaum *et al*, 1998; Lammich *et al*, 1999; Jorissen *et al*, 2010; Kuhn *et al*, 2010). Taken together, SARS-CoV-2 infection of A549-ACE2 cells is facilitated by one or several metalloproteases.

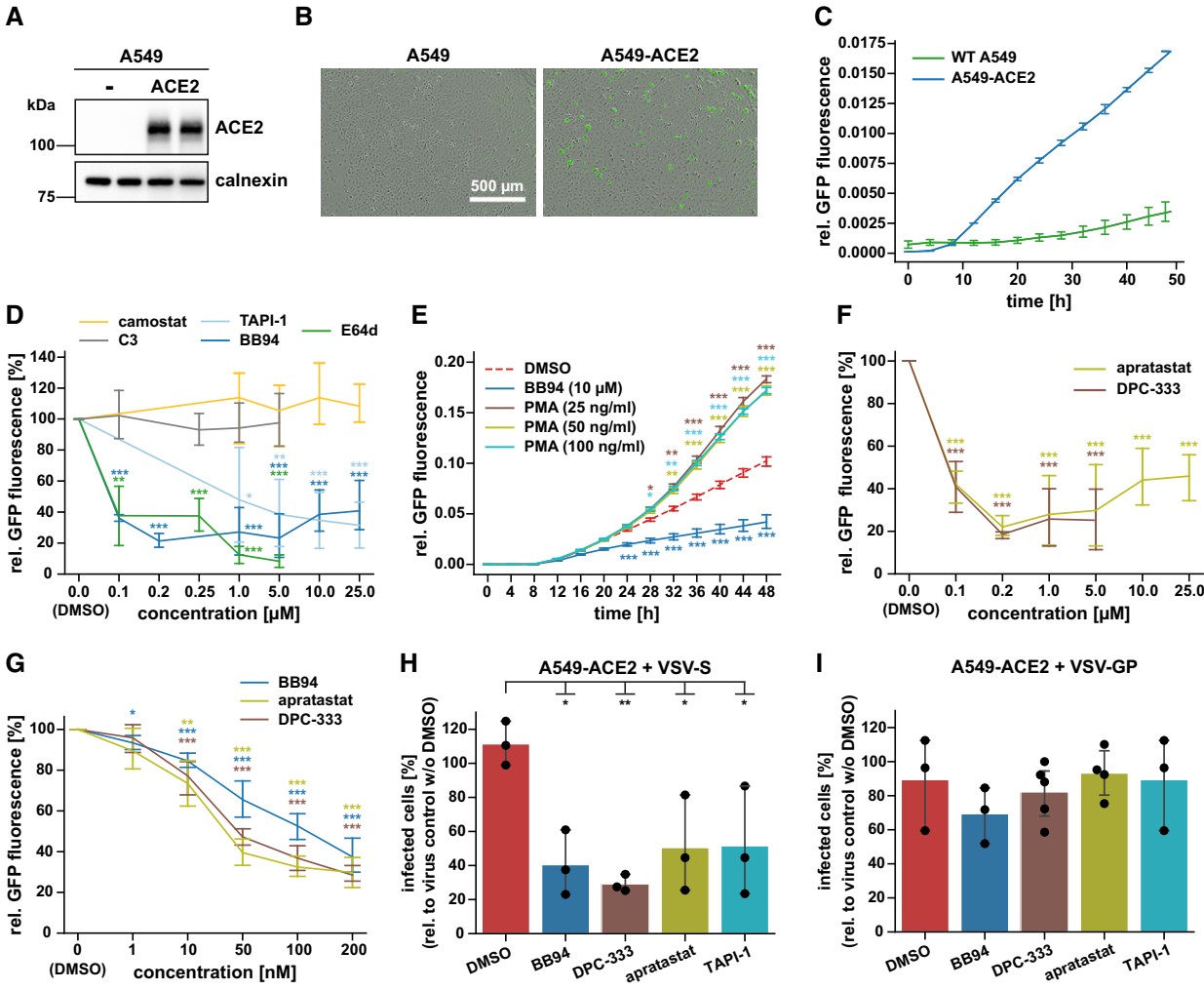

**Figure 1. Metalloprotease inhibitors reduce SARS-CoV-2 infection *in vitro*.**

A   Immunoblot analysis of ACE2 expression in the lysate of human lung A549 cells overexpressing ACE2 compared with nontransduced (−) A549 cells. Calnexin served as loading control.

B, C   Illustration of the Incucyte readout. Infection of A549- or ACE2-expressing A549 cells with SARS-CoV-2 encoding a GFP expression cassette. Infection of cells was monitored microscopically as green fluorescence through live-cell imaging at 24 h postinfection (hpi) (B) or during the time course of 48 h and shown as the mean of the GFP-positive area relative to the whole area covered by cells in the same well (C). Error bars represent mean ± SD derived from biological triplicates (*N* = 3). Scale bar, 500 µm.

D   A549-ACE2 cells were treated for 6 h with different concentrations of indicated inhibitors and then infected with SARS-CoV-2-GFP. GFP fluorescence relative to DMSO-treated control cells was determined 48 hpi as described in (C). Two-sided independent Student's *t*-test with Benjamini–Hochberg FDR correction was performed for every indicated concentration compared with DMSO. Asterisks indicate significance compared with DMSO (*N* = 3–6).

E   A549-ACE2 cells were treated for 30 min with PMA at indicated concentrations and then infected with SARS-CoV-2-GFP. Infection of cells was monitored as in (C). Two-way ANOVA with the Tukey's *post-hoc* multiple comparison test. Asterisks indicate significance compared with the DMSO control of each time point (*N* = 9).

F, G   A549-ACE2 cells were treated with apratastat, DPC-333, or BB94 at indicated concentrations. Experiment was conducted and analyzed as in (D). Two-sided independent Student's *t*-test with Benjamini–Hochberg FDR correction was performed for every indicated concentration compared with DMSO (0 µM). Asterisks indicate significance compared with DMSO (*N* = 3–6).

H, I   A549-ACE2 cells were treated for 1 h with indicated inhibitors (10 µM) or DMSO before inoculation with GFP-encoding spike pseudoparticles (VSV pseudotyped with SARS-CoV-2 spike protein (VSV-S)) (H) or as a control LCMV glycoprotein (VSV-GP) (I). At 16 hpi, infection was analyzed by counting GFP-positive cells. Data are normalized to the number of infected cells in virus-only control wells without DMSO (*N* = 3–5). One-way ANOVA with the Tukey's *post-hoc* multiple comparison test. Asterisks indicate significance compared with DMSO.

Data information: Data from (D–I) are represented as mean ± 95% CI from at least three independent experiments. **P* < 0.05, ***P* < 0.01, ****P* < 0.001.
Source data are available online for this figure.

## Metalloproteases contribute to infection with pseudotyped VSV

We used a replication-deficient vesicular stomatitis virus pseudotyped with the SARS-CoV-2 S protein (VSV-S), which expressed GFP and is a commonly used virus model to study the SARS-CoV-2 entry process (Hoffmann *et al*, 2020b; Riepler *et al*, 2020). Similar to SARS-CoV-2 (Fig 1D–G), the infection with VSV-S was inhibited by 50–60% by the metalloprotease inhibitors BB94, TAPI-1, apratastat,

and DPC-333 (Fig 1H). As a negative control, we used VSV pseudo-typed with the glycoprotein (VSV-GP) of lymphocytic choriomeningitis virus (LCMV) that is unrelated to the SARS-CoV-2 S protein and uses α-dystroglycan, but not ACE2, as the host cell receptor (Cao et al, 1998). While this virus also infected A549-ACE2 cells, the infection was not significantly affected by metalloprotease inhibitors (Fig 1I). We conclude that the metalloprotease-dependent viral entry into A549-ACE2 cells depends on the SARS-CoV-2 S protein and is observed for both SARS-CoV-2 and spike-pseudotyped particles.

## Metalloproteases facilitate infection with SARS-CoV-2 variants of concern

Next, we tested whether the metalloprotease inhibitors also impaired infection of A549-ACE2 cells with clinical isolates of SARS-CoV-2 (SARS-CoV-2-MUC-IMB-1 and Innsbruck 1.2 isolate) and whether they would also reduce infection with the new, more rapidly spreading alpha (B.1.1.7), beta (B.1.351), delta (B1.617.2) and omicron (B.1.1.529) variants of SARS-CoV-2 (preprint: Tegally et al, 2020; Thi Nhu Thao et al, 2020). Because the original viral isolates without a GFP expression cassette were used, infection was measured by RT-qPCR-mediated quantification of viral RNA being replicated in infected A549-ACE2 cells. Indeed, BB94 treatment potently reduced viral RNA levels after infection with all SARS-CoV-2 isolates tested, including the alpha, beta, delta, and omicron virus variants (Fig 2A and B). Similar results were obtained for the clinically tested metalloprotease inhibitor DPC-333, which inhibited infection of A549-ACE2 cells with the Innsbruck 1.2 isolate virus and the beta and delta virus variant to a similar extent as BB94 (Fig 2B). Reduced viral RNA levels were observed in a dose- and time-dependent manner for BB94 treatment (Fig EV1G and H). The inhibitory effect of BB94 was also observed when using a plaque assay as a readout to quantify the production of infectious particles and was similar to the effect of the lysosomal inhibitor E64d (Fig EV1I). These experiments underline the relevance of a metalloprotease for infection by SARS-CoV-2 and its variants of concern.

## Metalloprotease inhibitor reduces viral load in primary human bronchial cells

With our finding that metalloprotease inhibitors are able to reduce SARS-CoV-2 infections in a human lung cell line, we next tested whether a similar inhibitory effect is also seen ex vivo in primary lung cells from healthy human donors. We used normal human bronchial epithelial cells (NHBE), which are heterogeneous and comprise different cell types that can be infected with SARS-CoV-2, both in vitro and in patients (Bao et al, 2020; Chua et al, 2020; Lukassen et al, 2020; Radzikowska et al, 2020). NHBEs express ADAM proteases and TMPRSS2 (Fig EV2A and B). To dissect the involvement of both proteases we treated NHBEs with either the metalloprotease inhibitor BB94 or the TMPRSS2 inhibitor camostat or both and infected the cells with SARS-CoV-2 to follow virus growth by RT-qPCR. Compared with control-treated cells, camostat reduced viral RNA levels by 70% (Fig 2C), demonstrating that TMPRSS2 facilitates viral entry in this primary human cell model. Notably, BB94 reduced viral RNA to a similar extent, demonstrating

an important role for both TMPRSS2 and metalloproteases for SARS-CoV-2 entry into primary human cells. A lower concentration of BB94 (1 μM) had a mild inhibitory effect, demonstrating concentration-dependent inhibition of infection ex vivo. Combined treatment with camostat and BB94 showed a trend to a further reduction than either of the drugs applied independently. Similar results were obtained when infecting NHBE cells from different donors with the GFP-expressing SARS-CoV-2 virus and using GFP fluorescence intensity as a readout (Fig 2D). DPC-333 and apratastat reduced GFP fluorescence intensity to a similar degree as BB94. In this dataset, more variability was observed for camostat and also for the lysosomal inhibitor E64d, which both reduced GFP expression by approximately 50%. Co-treatment with BB94 and either camostat or E64d or cotreatment with all three protease inhibitors (BB94, camostat, and E64d) together efficiently reduced GFP expression (Fig 2D), demonstrating the need for proteolytic cleavage for SARS-CoV-2 infection. Similar results were obtained for A549-ACE2 cells upon cotreatment with BB94 and E64d and upon infection with the GFP-expressing SARS-CoV-2 (Fig EV2C).

Postinfection drug addition resembles more closely a therapeutic situation in patients. To test whether the inhibitors are also active in postinfection scenarios, we added them 4 h after viral infection (postinfection) and monitored SARS-CoV-2-GFP growth. For both NHBE and A549-ACE2 cells, the postinfection treatment with protease inhibitors had similar effects as compared to the pretreatment drug addition but showed greater variability among the donors, so that not all treatments resulted in significant changes (Figs 2E and EV2D). In conclusion, metalloprotease inhibitors that block ADAMs are effective in reducing viral load in primary human lung cells and are effective in postexposure scenarios.

## ADAM10 and ADAM17 facilitate SARS-CoV-2 infections

Among the metalloproteases blocked by BB94, TAPI-1, apratastat, and DPC-333, ADAM10 and ADAM17 are the major shedding enzymes (Moss et al, 2008; Lichtenthaler et al, 2018). To test whether ADAM10 and/or ADAM17 may be required for SARS-CoV-2 infection of A549-ACE2 cells, we used CRISPR/Cas9 and generated A549-ACE2 cells with a single knock-out of ADAM10 and ADAM17 or with a double knock-out of both proteases. For each protease, two distinct gRNAs were used. Knock-out of ADAM10 and ADAM17 was verified by cell lysate immunoblots compared with A549-ACE2 cells expressing a non-targeting control (NTC) gRNA. For both ADAM10 and ADAM17, the gRNAs (KO1, KO2) efficiently knocked out the proteolytically inactive, immature precursor form of the protease and the mature, fully processed, and proteolytically active protease form (Fig 3A and B). For KO2 a low amount of the immature protease form of both proteases remained (indicated with *), but KO2 corresponds to a full loss of function, as seen with the loss of sAPPα release from ADAM10 KO cells (Fig EV3A). Cellular ACE2 levels were not altered in the ADAM10 and ADAM17 KO cells but showed a trend of an increase in the double knock-out cells (Fig 3A). Noteworthy, knock-out of ADAM10 increased the abundance of the mature, active ADAM17, and, conversely, knock-out of ADAM17 increased levels of mature, active ADAM10 (Fig 3A and quantifications in B), suggesting compensatory changes among these two homologous ADAM proteases. In the A549-ACE2 double knock-out (ADAM10/17 dKO) cells, both proteases were efficiently depleted, as

seen by the absence of their immature and mature forms (Fig 3A) and the lack of sAPPα release (Fig EV3A).

Knock-out and NTC cells were infected with wild-type SARS-CoV-2 isolate. Infection was measured by quantifying viral RNA produced in infected cells. To determine whether the protease

knock-out would mimic the effect of the metalloprotease inhibitors, cells were additionally treated with BB94 or DMSO as control. In agreement with the results in Fig 1D, BB94 treatment strongly reduced infection by about 70% in NTC cells (Fig 3C). ADAM10 KO1 showed a mild and variable reduction of infection (Fig 3C), but

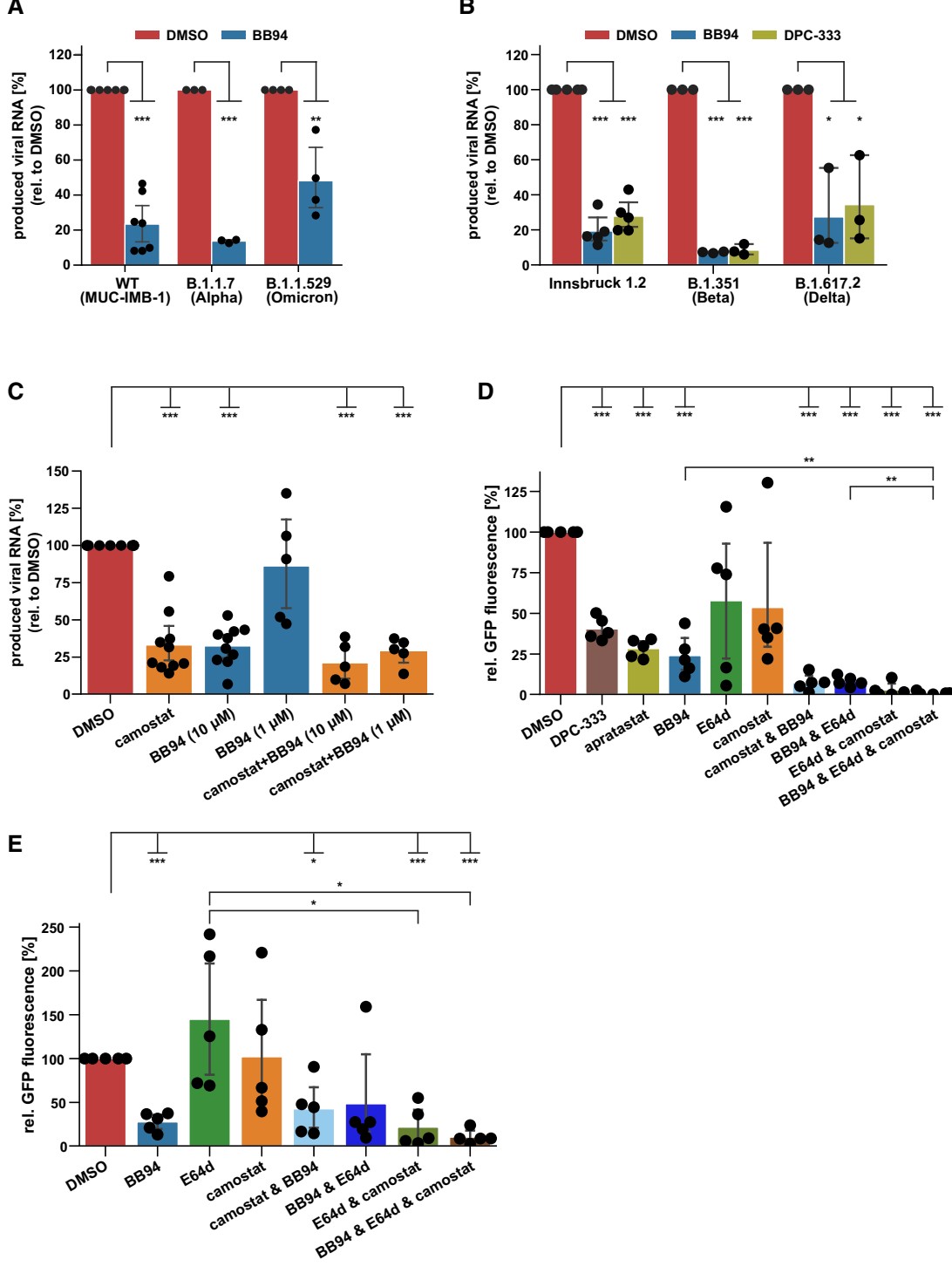

**Figure 2.**

**Figure 2. Metalloprotease inhibitors reduce viral infectivity of SARS-CoV-2 variants and in primary cells *ex vivo*.**

A, B    RT-qPCR quantification of viral RNA (SARS-CoV-2 N gene relative to *RPLP0*) upon infection of A549-ACE2 cells with indicated SARS-CoV-2 strains at 24 hpi. Cells were pretreated with BB94 (10 μM), DPC-333 (10 μM), or DMSO for 6 h. Data are normalized to DMSO (N = 3–7). A two-sided independent Student's *t*-test with Benjamini–Hochberg FDR correction was performed.

C    Normal human bronchial epithelial cells (NHBE) from ten donors were infected with SARS-CoV-2. NHBE cells were pretreated with camostat mesylate (10 μM), BB94 (1 or 10 μM), or DMSO for 6 h. RNA was isolated at 24 hpi, and levels of SARS-CoV-2 N relative to *RPLP0* from total cellular RNA were measured by RT-qPCR. Data are normalized to the DMSO control (N = 5–10). Two-sided independent Student's *t*-test with Benjamini–Hochberg FDR correction.

D, E    NHBE cells from five donors were infected with GFP-expressing SARS-CoV-2. Infection of cells was monitored microscopically as green fluorescence through live-cell imaging at 72 h postinfection (hpi) and is shown as the mean of the GFP-positive area relative to the whole area covered by cells in the same well. In (D), NHBE cells were pretreated with the indicated drugs for 6 h before infection or were treated in (E) with the indicated drugs 4 h after infection. Data are normalized to the DMSO control (N = 5). Two-sided independent Student's *t*-test with Benjamini–Hochberg FDR correction.

Data information: Data from (A–E) are represented as mean ± 95% CI from at least three (A, B) or five/ten biological replicates (C–E). *P < 0.05, **P < 0.01, ***P < 0.001. See also Fig EV2.

Source data are available online for this figure.

this was not seen for ADAM10 KO2 cells, demonstrating that ADAM10-deficiency alone does not have a major impact on A549-ACE2 infection. For both ADAM10 knock-out cell lines, BB94 treatment further reduced infection to the level of the treated NTC control cells (Fig 3C). In contrast to ADAM10, both single knock-outs of ADAM17 reduced infection by 40–50%, and this rate was further reduced by the addition of BB94. Double knock-out (ADAM10/17 KO) cells lacking both ADAM10 and ADAM17 showed the strongest reduction of infection similar to BB94-treated NTC cells, and this double knock-out effect was not significantly further reduced with additional BB94 treatment (Fig 3C). Together, the data demonstrate that the single knock-out of ADAM17 is sufficient to reduce infection, but that knock-out of both ADAM17 and ADAM10 is required to fully mimic the effect of a metalloprotease inhibitor, potentially due to compensatory regulation of both proteases, which was also seen in the immunoblot analysis (Fig 3A and B). Because A594-ACE2 cells do not express TMPRSS2, the involvement of ADAM17 and potentially ADAM10 in SARS-CoV-2 infection must be TMPRSS2-independent in this assay.

## Lung expression of ADAM10 and ADAM17 correlates with COVID-19 severity

Having established the role of ADAM10 and ADAM17 in SARS-CoV-2 infection, we tested whether expression of both proteases was detected endogenously in single-cell transcriptomes of healthy human lungs and bronchoalveolar lavage (BAL) fluid of SARS-CoV-2 infected individuals. Using published single-cell RNA sequencing (scRNA-seq) datasets (Han *et al*, 2020a; Data ref: Han *et al*, 2020b; Liao *et al*, 2020a; Data ref: Liao *et al*, 2020b), ADAM10 and ADAM17 were found to be widely expressed in healthy lung cells, including AT1 and AT2 pneumocytes and in macrophages (Fig EV3B and C). The protease TMPRSS2, which can prime the S protein and enhance infectivity (Hoffmann *et al*, 2020b; Matsuyama *et al*, 2020), was also expressed in different lung cell types, but more concentrated in pneumocytes (including AT1 and AT2 cells) and less expressed in other lung cell types, including macrophages (Fig EV3B and C). The scRNA-seq analysis of control and COVID-19 patient BAL fluid provides information on different lung cell types (immune cells, epithelial cells) and identifies infected cells by detection of viral RNA in single-cell transcriptomes (Liao *et al*, 2020a; Data ref: Liao *et al*, 2020b; Fig 3D). Most of the infected cells were found to express ADAM10 and/or ADAM17, while few cells

expressed TMPRSS2 (Fig 3E–G). Notably, expression of ADAM10 and ADAM17, and also of TMPRSS2 and ACE2 was higher in the severe COVID-19 patients compared with healthy controls (Fig 3F). We conclude that both ADAM10 and ADAM17 are widely expressed in lung cells, including those that are infected with SARS-CoV-2.

## Inhibition of ADAM proteases reduces lung cell fusion

Syncytia formation of pneumocytes with frequent multinucleation is a pathological hallmark in the lung of COVID-19 patients and is assumed to occur through the fusion of cells expressing the S protein and its receptor ACE2 (Buchrieser *et al*, 2020; Bussani *et al*, 2020; Braga *et al*, 2021). Both S and ACE2 are also required for lung cell infection (Fig 1), raising the possibility that syncytia formation may also depend on ADAM10 and/or ADAM17 and may be blocked by metalloprotease inhibitors. To test this hypothesis, A549-ACE2 NTC cells were co-cultured with HEK293T cells transiently expressing both S and GFP (Fig 4A), which resulted in large, GFP-positive syncytia (Fig 4B). Notably, the addition of BB94 reduced the GFP-positive syncytia area by more than 50% and also the number of nuclei contained in each syncytium (Fig 4B and C). The single knock-out of ADAM10 in A549-ACE2 cells was sufficient to reduce syncytia formation and the number of nuclei in syncytia to the level of BB94-treated NTC cells. In contrast, the single knock-out of ADAM17 did not affect syncytia formation, and the double knock-out of ADAM10 and ADAM17 had a similar effect as the single ADAM10 knock-out alone (Fig 4B and C). Moreover, the lysosomal inhibitor E64d, which efficiently blocked SARS-CoV-2 cell infection (Fig 1), did not block cell fusion (Fig EV4). This data reveals that—in contrast to lung cell infection, which required ADAM17 and lysosomal proteases—syncytia formation depends on ADAM10 and does not require lysosomal proteases. Thus, syncytia formation and cell infection, which both occur in COVID-19 pathogenesis, are controlled by different sets of proteins, even though both steps require the expression of ACE2 and S.

## ADAM10 and ADAM17 cleave the spike protein

To determine which protease substrate may be relevant for ADAM17-dependent SARS-CoV-2 cell entry and ADAM10-dependent lung cell fusion, we considered the SARS-CoV-2 S protein because both processes are facilitated upon proteolytic priming of S. TMPRSS2 may enhance SARS-CoV-2 infection of cells through

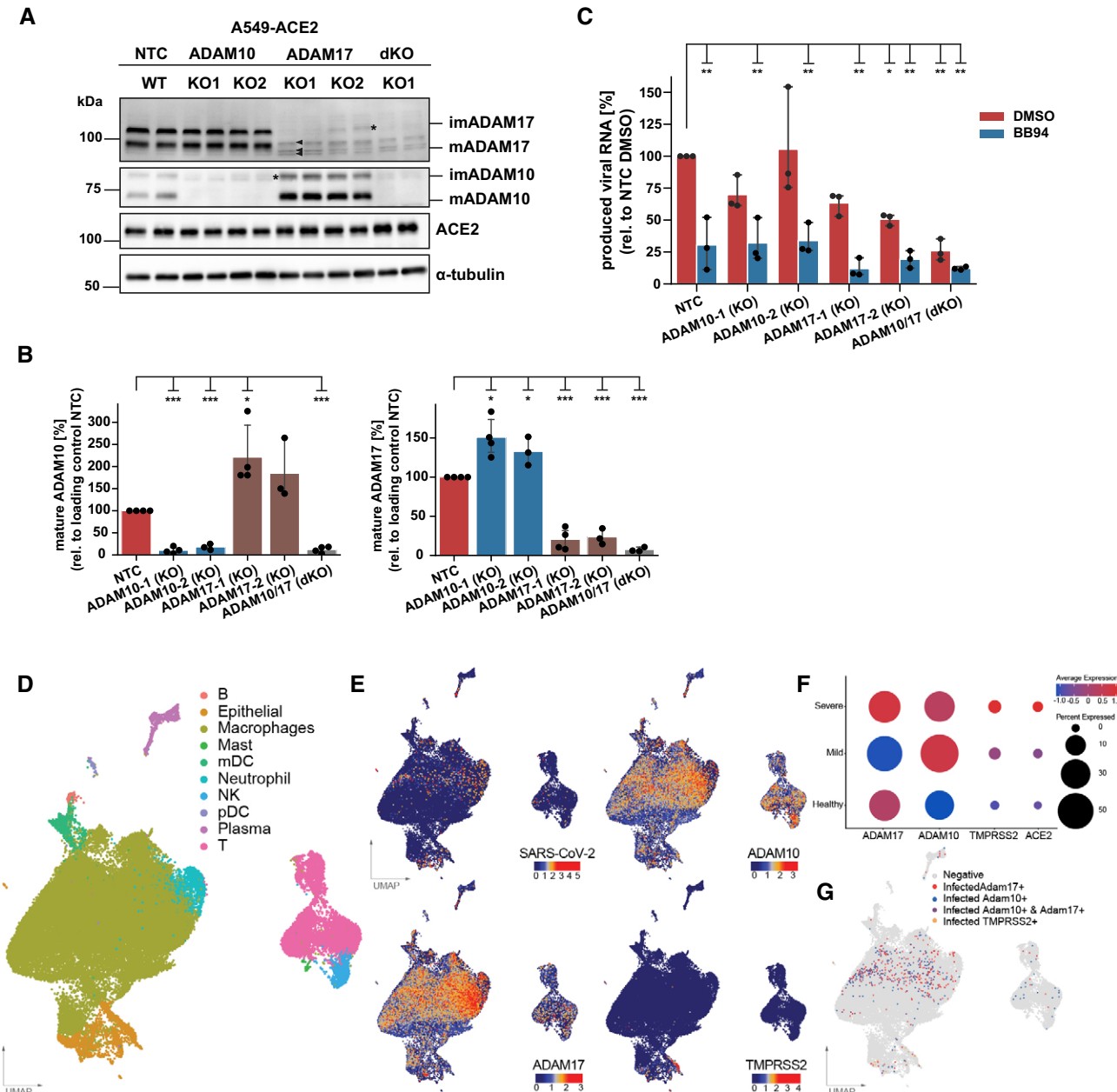

**Figure 3. ADAM17 facilitates SARS-CoV-2 infection.**

A Western Blot (WB) analysis of A549-ACE2 knock-out (KO) lines. Two different KO lines were generated using distinct CRISPR guide RNAs for both ADAM10 and ADAM17. A non-targeting control (NTC) guide RNA served as control. Shown are two replicates for each cell line. Mature (m) and immature (im) forms of ADAM10 and ADAM17 are indicated. ACE2 levels remained unaffected. α-tubulin served as loading control. *: remaining, low amount of immature ADAM10 and ADAM17. The double KO (dKO) was generated from guide 1 of each ADAM10 and ADAM17.

B Quantification of the abundance of mature ADAM10 and ADAM17 protein forms from (A). Data are normalized to loading control α-tubulin and to NTC ($N = 3$–4). The band of mADAM17 runs very close to three unspecific bands (marked with arrowheads)—one closely above and two closely below mADAM17 (A). Thus, the quantification of remaining mADAM17 in the ADAM10 KO includes also these three unspecific bands. Two-sided independent Student's $t$-test with Benjamini– Hochberg FDR correction.

C RT-qPCR quantification of viral RNA (SARS-CoV-2 N gene relative to *RPLP0)* upon infection of A549-ACE2 NTC or KO cells with SARS-CoV-2. Cells were treated with BB94 (10 μM) or DMSO for 6 h prior to infection and RNA was isolated at 24 hpi. Two-way ANOVA with the Tukey's *post hoc* multiple comparison test. Data are normalized to NTC DMSO ($N = 3$).

D–G UMAP of 63,103 cells from BAL fluid derived from infected individuals (Liao *et al*, 2020). (D) Individual cell types are indicated in color. (E) Normalized counts of SARS-CoV-2 reads, ADAM10, ADAM17, and TMPRSS2 gene expression levels. (F) Dot plot of ADAM10, ADAM17, and TMPRSS2 expression, grouped by disease severity. Expression levels are color-coded, and the percentage of cells expressing the aforementioned genes is size-coded. (G) Cells that are infected and additionally express ADAM17 (ADAM17+, red), ADAM10 (ADAM10+, blue), both ADAMs (ADAM10+ ADAM17+, purple), or TMPRSS2 (TMPRSS2+, orange) are indicated. See also Fig EV3.

Data information: Data from (B, C) are represented as mean ± 95% CI from at least three independent experiments. *$P < 0.05$, **$P < 0.01$, ***$P < 0.001$.
Source data are available online for this figure.

cleavage of S within the S2 subunit at a site referred to as S2' site (Millet & Whittaker, 2018; preprint: Essalmani *et al*, 2020; Hoffmann *et al*, 2020a; Nguyen *et al*, 2020). Other, as yet unknown proteases are expected to also contribute because SARS-CoV-2 also infects

cells that do not express TMPRSS2, such as A549-ACE2, and as seen in the single-cell RNA-seq data analysis (Fig 3E and G). Likewise, syncytia formation is enhanced upon the addition of exogenous proteases, such as TMPRSS2 or trypsin, presumably through S protein

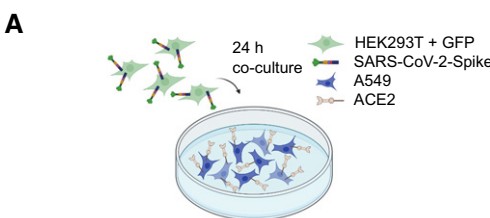

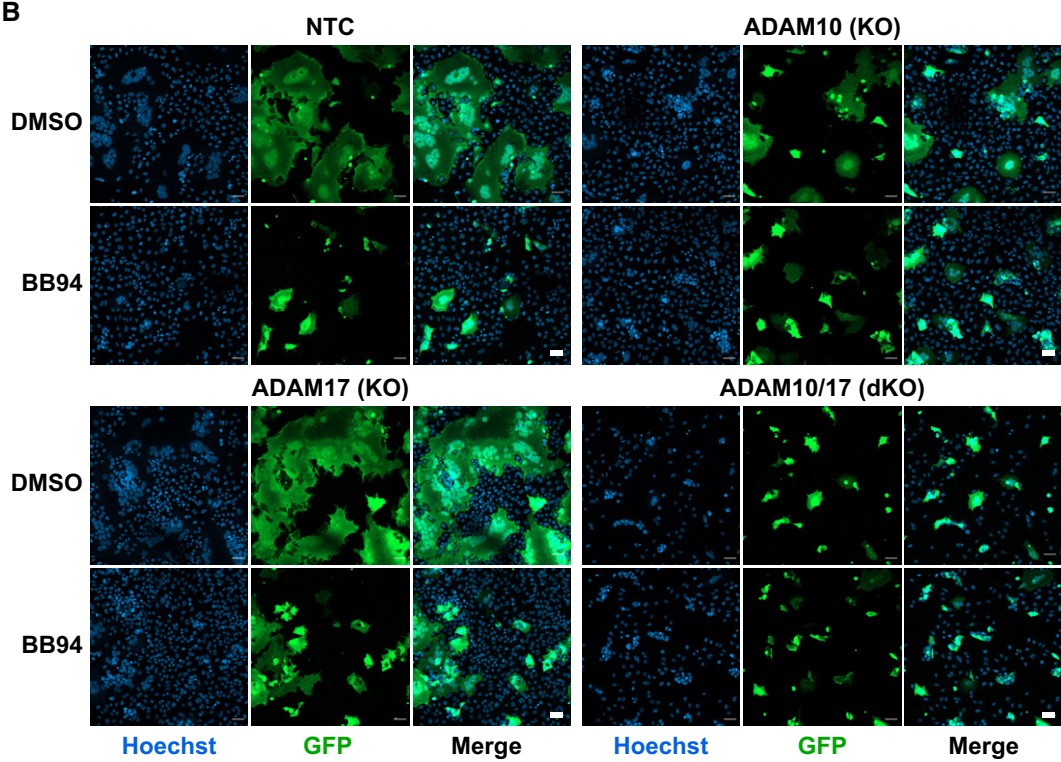

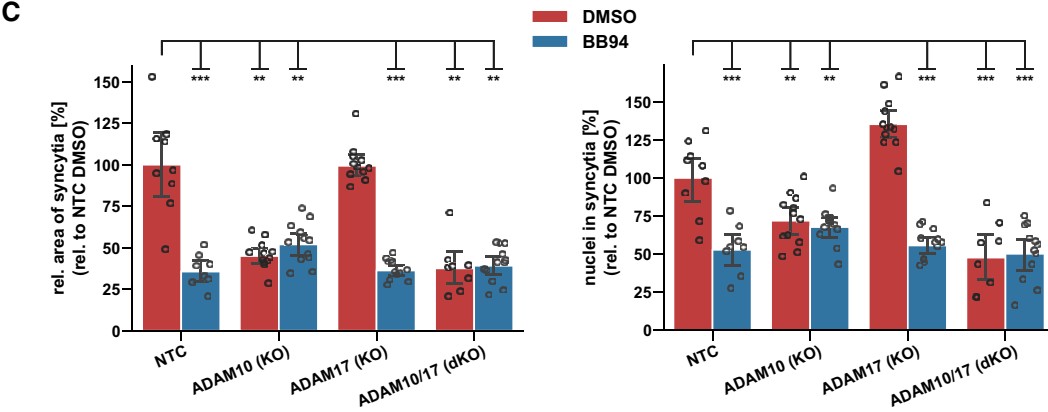

**Figure 4.**

◀

**Figure 4.  ADAM10 is required for lung cell syncytia formation.**

A   Experimental scheme of syncytia formation assay. Donor cells (HEK293T) co-expressing the SARS-CoV-2 spike protein along with GFP were co-cultured with A549-ACE2 NTC or KO acceptor cells. After 24 h, co-cultures were analyzed by confocal microscopy.
B   Representative images of syncytia formation assay. Large syncytia were observed for NTC and ADAM17 KO cells but were strongly reduced when ADAM10 was knocked out, either alone or together with ADAM17 or upon inhibition with BB94 (10 μM). For both ADAMs, the cell lines obtained with gRNA sequence 1 were used. Images show GFP fluorescence and Hoechst staining. Scale bars, 50 μm.
C   Quantification of images from fluorescence microscopy (B). Left plot: the area of the fused cells in green (GFP) was quantified and normalized to the Hoechst signal (blue) of the entire image to account for differences in cell density. Right plot: The nuclei within syncytia were determined by calculating the ratio between the Hoechst signal within syncytia and the total Hoechst signal in the entire image. The data are normalized to NTC DMSO ($N \geq 8$). Two-way ANOVA with Tukey's correction for multiple comparisons. All data are represented as mean $\pm$ 95% CI of at least three independent experiments. **$P < 0.01$, ***$P < 0.001$. See also Fig EV4.

Source data are available online for this figure.

cleavage (Nguyen *et al*, 2020; Hörnich *et al*, 2021). Thus, we tested whether recombinant ADAM10 and ADAM17 may prime the S protein similar to TMPRSS2. We used recombinant, C-terminally His-tagged S protein *in vitro* and observed a molecular weight shift from 70 to 55 kDa upon addition of either ADAM10 or ADAM17, similar to what was seen in the presence of TMPRSS2 (Fig 5A and B). Interestingly, for cleavage by ADAM10 several S2′ fragments of similar molecular weight were observed, raising the possibility of multiple cleavage sites around the known S2′ site. Together, this data demonstrates that ADAM10 and ADAM17 are able to cleave the S protein similar to TMPRSS2 in a cell-free assay.

To determine whether an interaction of ADAM17 and S can be seen in cells, we used a proximity ligation assay (PLA) in A549-ACE2 cells, in which the colocalization of antibodies bound to ADAM17 and S was measured. While the use of single antibodies to either ADAM17 or S as expected did not generate a PLA signal and served as a negative control, a PLA signal was clearly detected when both antibodies were used (Fig 5C). After infection, cells were placed for 2 h at 4°C to reduce endocytosis and then moved to 37°C to allow endocytosis or kept at 4°C. The PLA signal was seen in cells kept at 4°C and was enhanced in cells at 37°C. We conclude that the interaction of S and ADAM17 may start at the plasma membrane and is enhanced when endocytosis is possible.

ADAM17, which facilitated SARS-CoV-2 cell infection (Fig 3C), is known to cleave the S receptor ACE2, which is a single-span transmembrane protein (Lambert *et al*, 2005; Jia *et al*, 2009). In agreement with the previous studies, BB94 and DPC-333 reduced soluble ACE2 ectodomain (sACE2) release from A549-ACE2 cells, but only by up to 50% (Fig 5D). Full-length ACE2 levels were not significantly altered (Fig EV5A and B). Although the reduction of sACE2 was clearly less than the degree of inhibition of viral infection (Fig 1F), we considered the possibility that ADAM17-mediated shedding of ACE2 and the resulting release of sACE2 may be an additional mechanism through which ADAM17 contributes to SARS-CoV-2 infection. It remains unclear at which peptide bond ACE2 is cleaved by ADAM17 (Jia *et al*, 2009; Heurich *et al*, 2014), and we were not successful in designing ACE2 mutants that abrogate ACE2 shedding to test for a role of ACE2 shedding in SARS-CoV-2 infection. Therefore, we used an alternative approach and tested whether recombinant sACE2—the ADAM17 shedding product of ACE2—affects A549-ACE2 infection with the spike pseudoparticles (VSV-S). We observed a dose-dependent inhibition of infection (Fig 5E), in agreement with previous studies using Vero cells and human organoids (Monteil *et al*, 2020; Hoffmann *et al*, 2021). This demonstrates that sACE2 is not required for infection but instead acts as an inhibitor of infection.

Thus, ADAM17 is unlikely to facilitate SARS-CoV-2 cell entry through cleavage of ACE2 shedding and sACE2 release.

## Discussion

Our study highlights two important new biological concepts for mechanistically understanding SARS-CoV-2 infections and COVID-19. The first concept is the identification of ADAM17 as a host factor that potentiates infectivity of SARS-CoV-2 in lung cells. Second, we demonstrate that ADAM10 is required for lung cell syncytia formation, which is a pathological hallmark of the lungs of COVID-19 patients.

The metalloproteases ADAM10 and ADAM17 have fundamental functions in the communication between cells, including in Notch signaling (ADAM10) and skin and intestinal barrier formation and maintenance (ADAM17; Pan & Rubin, 1997; Peschon *et al*, 1998; Hartmann *et al*, 2002). Both proteases are also linked to pathophysiological conditions, such as Alzheimer's disease (ADAM10) and inflammation (ADAM17; Peschon *et al*, 1998; Lammich *et al*, 1999; Jorissen *et al*, 2010; Kuhn *et al*, 2010). ADAM10 and ADAM17 are the closest homologs within the ADAM protease family. Both shed numerous membrane protein substrates, some of which can be cleaved by both (Dreymueller *et al*, 2015; Zunke & Rose-John, 2017; Lichtenthaler *et al*, 2018). Yet, they also have distinct physiological functions, which is illustrated by distinct phenotypes of the respective knock-out mice that show embryonic (ADAM10) and perinatal (ADAM17) lethality (Peschon *et al*, 1998; Hartmann *et al*, 2002).

Our study identifies a new but distinct function for both ADAM proteases in COVID-19 pathogenesis. Using pharmacological inhibition and genetic ablation, we establish ADAM17 as a host factor required for efficient SARS-CoV-2 infection of human lung cells. Moreover, metalloprotease inhibitors that block ADAM17 reduced SARS-CoV-2 infection in primary human lung cells. ADAM17 may also contribute to the infectivity of another betacoronavirus, the SARS-CoV, but this remains controversially discussed (Lambert *et al*, 2005; Haga *et al*, 2008, 2010; Glowacka *et al*, 2010; Heurich *et al*, 2014).

For ADAM10, our results demonstrate a function at a later stage in COVID-19 pathogenesis, i.e., in syncytia formation of lung cells, which is one pathological hallmark in the lungs of COVID-19 patients and assumed to occur through the fusion of infected cells expressing the S protein with cells expressing its receptor ACE2 (Bussani *et al*, 2020; Braga *et al*, 2021). This syncytia formation is also seen *in vitro*, as in our study and in previous ones (e.g.,

Buchrieser *et al*, 2020; Ou *et al*, 2020). While none of the previous studies demonstrated a requirement for ADAM10 or other ADAM proteases, it is interesting to note that some of them found the cell fusion to be inhibited with broad-spectrum metalloprotease inhibitors that can also block ADAM10 (Nguyen *et al*, 2020; Hörnich *et al*, 2021), or to be promoted with ionomycin (Braga *et al*, 2021), a calcium ionophore that is also used to activate cleavage of ADAM10

substrates (Horiuchi *et al*, 2007; Le Gall *et al*, 2009). These findings are in line with our identification of ADAM10 as being essential for efficient cell fusion.

Although we find that ADAM10 and ADAM17 function at different steps in COVID-19 pathogenesis, we provide evidence that both proteases act through proteolytic priming of the S protein. S is a furin-processed heterodimer consisting of an S1 and an S2 subunit

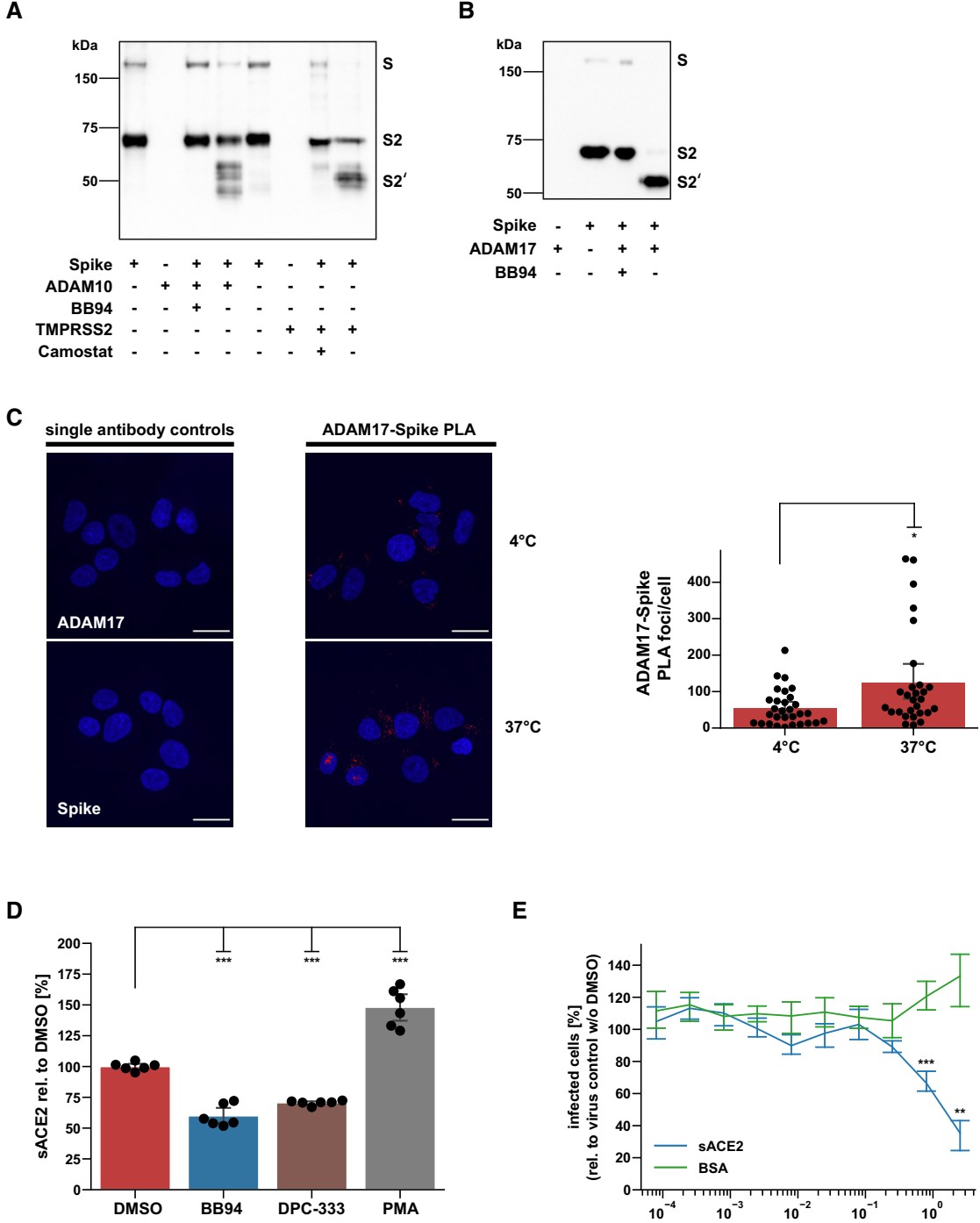

**Figure 5.**

◄

**Figure 5.  ADAM10 and ADAM17 contribute to spike protein priming and ACE2 cleavage, while soluble ACE2 prevents infection.**

A, B    *In vitro* cleavage assay of SARS-CoV-2 spike protein. Recombinant spike protein (1 μg) was digested with either ADAM10, TMPRSS2 (A), or ADAM17 (B) for 24 h at a 1:1 ratio and analyzed by Western Blot. Camostat mesylate or BB94 (10 μM) were preincubated with TMPRSS2 or ADAM proteases, respectively, for 15 min before the addition of spike protein. The spike protein and fragments were detected using an antibody against its C-terminal 6xHIS tag.

C    Representative images and quantification of a proximity ligation assay (PLA) between the SARS-CoV-2 spike protein (Spike) and ADAM17 in A549-ACE2 cells. Cells were infected with SARS-CoV-2 and placed on ice (4°C). After 2 h, cells were shifted to 37°C for 15 min or left on ice. Images are presented as overlay between the nuclei (Hoechst, blue) and the PLA (pseudocolor: red) channel. Scale bar, 20 μm. The number of PLA foci per cell was determined for the acquired images (N = 28–30). A two-sided independent Student's *t*-test with Benjamini–Hochberg FDR correction was performed.

D    A549-ACE2 cells were treated for 48 h with BB94 (10 μM), DPC-333 (5 μM), or DMSO. PMA (25 ng/ml) was added for 3 h before harvest to stimulate ACE2 shedding. Concentration of the soluble ACE2 fragment was determined in the conditioned medium by ELISA and is shown relative to the concentration in the DMSO control (N = 6). One-way ANOVA with the Tukey's *post-hoc* multiple comparison test.

E    GFP-encoding spike pseudoparticles (VSV) pseudotyped with SARS-CoV-2 spike protein (VSV-S) were incubated with indicated amounts of recombinant, soluble ACE2 (sACE2), or bovine serum albumin (BSA) as a control prior to infection of A549-ACE2 cells. At 16 hpi, infection was analyzed by counting GFP-positive cells. Data are normalized to the number of infected cells in virus-only control wells without sACE2 or BSA (N = 3). Two-sided independent Student's *t*-test with Benjamini–Hochberg FDR correction.

Data information: Data from (C–E) are represented as mean ± 95% CI of at least three independent experiments. *$P < 0.05$, **$P < 0.01$, ***$P < 0.001$.
Source data are available online for this figure.

(Walls *et al*, 2020). After binding of the SARS-CoV-2 virus to ACE2 on host cells, additional proteolytic priming at the S2' site is required that occurs within the S2 domain just N-terminal of the fusion peptide that mediates fusion of the virus with the host cell membrane (Coutard *et al*, 2020; Hoffmann *et al*, 2020b). Likewise, syncytia formation between cells expressing ACE2 and cells expressing S is facilitated through proteolytic priming of S (Hoffmann *et al*, 2018, 2020a). One protease, that is able to cleave at the S2' site, is TMPRSS2 (Matsuyama *et al*, 2020; Zang *et al*, 2020). Other proteases must also exist because: (i) cells without TMPRSS2 expression, such as A549-ACE2, are also efficiently infected with SARS-CoV-2, as shown in our study and in the previous ones (Stukalov *et al*, 2021; Zhu *et al*, 2021), and (ii) because little or no TMPRSS2 expression was detected in SARS-CoV-2-infected BAL fluid cells by scRNA-seq analysis. We demonstrate that ADAM10 and ADAM17 can also prime at or near to the S2' site because both proteases converted the S2 subunit of recombinant S protein *in vitro* to a similarly-sized S2' fragment as TMPRSS2. Such a function for ADAMs in S priming is in line with their broad expression in healthy human lungs and in SARS-CoV-2-infected BAL fluid cells from COVID-19 patients, where ADAM expression correlated with disease severity, as seen upon analysis of the scRNA-seq data. We conclude that different proteases, including TMPRSS2, ADAM10, ADAM17, and potentially others, may prime S2 and, thus, facilitate: (i) infection through the fusion of the SARS-CoV-2 virus with host cells, or (ii) syncytia formation of S-expressing SARS-CoV-2-infected cells with host cells.

The use of the specific protease in the viral or cell fusion event may possibly depend on their individual expression profile in each relevant cell type. If several of these proteases are expressed within one cell type, such as the A549-ACE2 cells in our study, the use of the individual protease may additionally be determined by the cellular location where the priming step occurs. In fact, S2 priming and viral fusion during infection can occur in two distinct cellular pathways, the utilization of which may strongly depend on the cell type used (Dittmar *et al*, 2021). One of them happens at the plasma membrane, in particular in TMPRSS2-expressing cells (preprint: Essalmani *et al*, 2020; Hoffmann *et al*, 2020a; Matsuyama *et al*, 2020; Shang *et al*, 2020). However, S2 priming and viral fusion can also occur in an alternative cell entry pathway that requires viral endocytosis and low pH and may be mediated by cathepsins to allow fusion with the

endo-lysosomal membrane (Hoffmann *et al*, 2020a; Ou *et al*, 2020; Shang *et al*, 2020). Given that the A549-ACE2 cells do not express TMPRSS2 and that their infection was not only reduced upon ADAM17 knock-out, but also with a lysosomal cathepsin inhibitor, we propose that ADAM17 acts in the lysosomal entry pathway, either at the plasma membrane just before endocytosis or after endocytosis (Fig 6A). In contrast to cell infection, the priming step and fusion of cells during syncytia formation occur at the plasma membrane (Fig 6B). This fusion step can be enhanced by TMPRSS2 and is the step that we found to be dependent on ADAM10 in the TMRPSS2-negative A549-ACE2 cells and independent of lysosomal cathepsins. Thus, we conclude that ADAM10 primes the S protein at the plasma membrane to allow syncytia formation (Fig 6B).

ADAM proteases typically cleave their substrates in cis when expressed within the same cell, but a cleavage of ephrin-A5 in trans was reported for ADAM10 (Janes *et al*, 2005). It is therefore conceivable that S protein priming by ADAM10 or ADAM17 may happen in cis or trans. If cleavage happens while the S1/S2 heterodimer is still bound to the receptor ACE2, the ADAM—being expressed on the same cell as ACE2—would cleave in trans (Fig 6). However, it is also possible that ADAM cleavage occurs after the insertion of the fusion peptide of the S2 subunit into the host cell membrane. In this case, the ADAM cleavage can be considered to be occurring in cis (Fig 6).

The results of our study suggest to test ADAM10 and ADAM17 as potential drug targets to reduce SARS-CoV-2 cell entry and lung cell syncytia formation. ADAM17 has been intensively studied as a drug target for inflammatory disorders. It is the protease that sheds various cytokines and their receptors from immune cells, including the cytokine TNFα (Black *et al*, 1997; Moss *et al*, 1997, 2008; Peschon *et al*, 1998; Horiuchi *et al*, 2007; Dreymueller *et al*, 2015; Zunke & Rose-John, 2017; Hsia *et al*, 2019; Calligaris *et al*, 2021), which is also increased during the cytokine storm in severe forms of COVID-19 and correlates with viral load in patients and with mortality in animal models (Catanzaro *et al*, 2020; Del Valle *et al*, 2020; Lucas *et al*, 2020; Karki *et al*, 2021). Two of the metalloprotease inhibitors tested in our study, DPC-333 and apratastat, were developed to block ADAM17-mediated TNFα shedding and previously entered phase 2 clinical trials for the treatment of the inflammatory joint disorder rheumatoid arthritis. Because of a lack of a disease-modifying

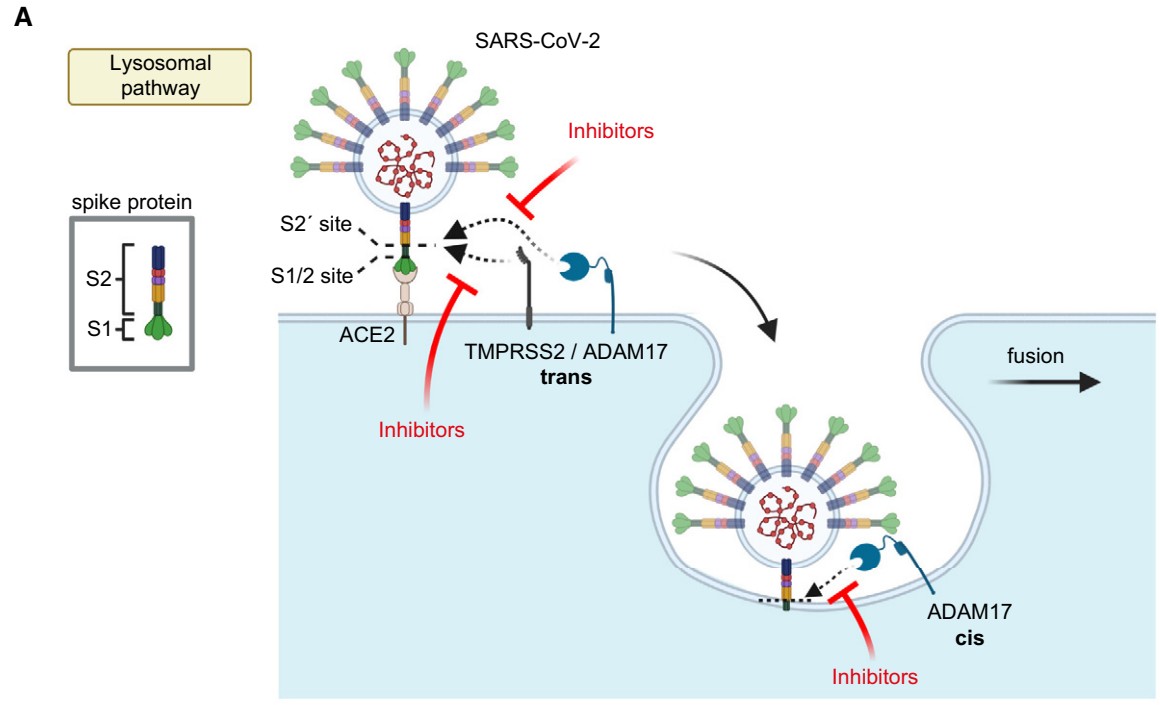

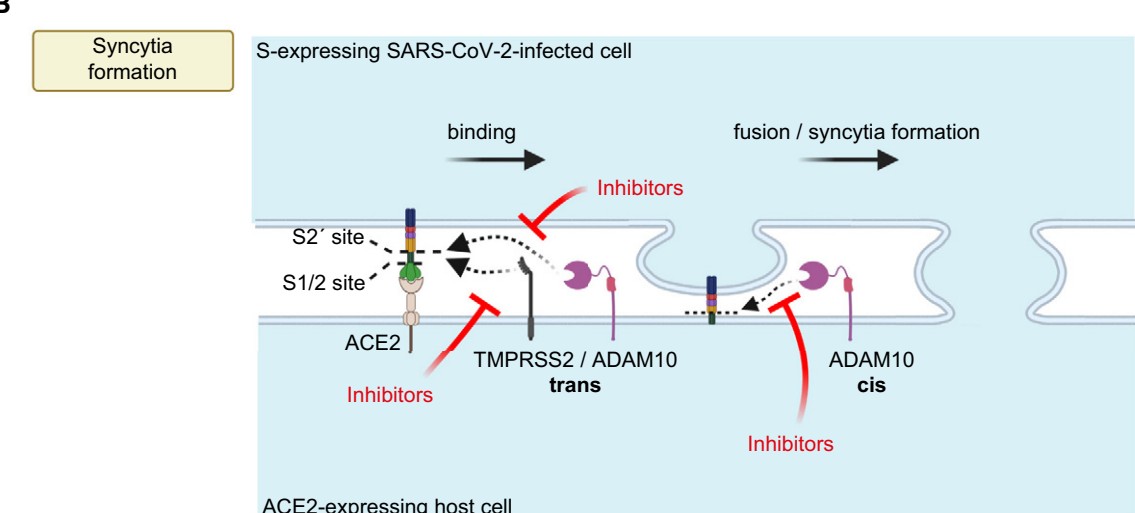

**Figure 6. Model for the role of ADAM proteases in SARS-CoV-2 viral entry and lung cell syncytia formation.**

A, B The SARS-CoV-2 spike protein S consists of the two subunits S1 and S2 and binds through S1 to ACE2 on host cells, which is required for viral fusion with host cells (A) and for syncytia formation (B). (A) TMPRSS2 may cleave S in trans and allow for the fusion of SARS-CoV-2 with the host cell plasma membrane. In contrast, ADAM17 may either cleave S in trans at the plasma membrane or may cleave it once the fusion peptide of S has inserted into the host plasma membrane after endocytosis, which is conceptually similar to a cleavage in cis. After ADAM17 cleavage, the fusion of SARS-CoV-2 with the host cell is likely to happen after endocytosis. Inhibition of ADAM17 reduces viral uptake and fusion. (B) For syncytia formation, the S protein expressed in an infected cell binds at the plasma membrane to ACE2 on another cell. This requires S priming by TMPRSS2 or ADAM10. The priming step may happen in trans, while SARS-CoV-2 is still bound to ACE2 or after insertion of the S2 subunit—which contains the fusion peptide—into the membrane of the ACE2-expressing cell, which may be seen as a cleavage in cis. Inhibitors of TMPRSS2 (e.g., camostat) or ADAM10 reduce cell fusion and syncytia formation. The figure was created with BioRender.

effect and concerns about a potential hepatotoxicity (DPC-333) and tendonitis (apratastat) upon chronic dosing, they were not further tested in phase 3 trials (Moss *et al*, 2008). ADAM17-targeting

inhibitors have short half-lives *in vivo* and are rapidly metabolized, in particular in rodent models (Qian *et al*, 2007; Shu *et al*, 2011). Thus, it remains to be seen whether sufficiently high drug

concentrations can be achieved in animal models of COVID-19 during the course of the disease.

Taken together, our study provides new concepts and insights into the mechanism of SARS-CoV-2 cell entry and syncytia formation and suggests ADAM10 and ADAM17 as potential targets for antiviral treatments for COVID-19.

## Materials and Methods

### Cell lines, reagents, and antibodies

A549 and VeroE6 cells and their respective culturing conditions were described previously (Stukalov et al, 2021). All cell lines were tested to be mycoplasma-free. C-terminally hemagglutinin (HA)-tagged ACE2 sequence was amplified from an ACE2 expression vector (kindly provided by S. Pöhlmann, Hoffmann et al, 2020b) into the lentiviral vector pWPI-puro. A549 cells were transduced twice, and ACE2-expressing A549 (A549-ACE2) cells were selected with puromycin (Stukalov et al, 2021). Lentivirus production, transduction of cells, and antibiotic selection were performed as described previously (Kuhn et al, 2010). RNA-isolation (Macherey-Nagel NucleoSpin RNA plus), reverse transcription (TaKaRa Bio PrimeScript RT with gDNA eraser), and RT-qPCR (Thermo Fisher Scientific PowerUp SYBR green) were performed as described previously (Stukalov et al, 2021).

The following antibodies were used: ADAM10 (Abcam-ab124695), ADAM17-cyto-GST (Schlöndorff et al, 2000), ACE2 (R&D Systems—AF933), alpha-tubulin (Abcam-ab4074), calnexin (Enzo-ADI-SPA-860), β-actin (Sigma-Aldrich—A5316), GFP (Fitzgerald-20R-GR011), anti-HIS (Clone 3D5, kind gift from Regina Feederle).

The following inhibitors were used: BACE inhibitor C3 (also known as BACE inhibitor IV) and TAPI-1 from Calbiochem; GI254023X, 1,10-phenanthroline, E64d, and apratastat from Sigma-Aldrich; batimastat (BB94) from APExBIO, DPC-333 was kindly provided to CPB by Robert Waltermire (Brystol-Myers Squibb, New Brunswick, NJ; Le Gall et al, 2010); the TMPRSS2 inhibitor camostat mesylate from Santa Cruz;

Other reagents: Lipofectamine 2000 and Polybrene from Thermo Fisher Scientific. PMA (Phorbol 12-myristate 13- acetate) from Sigma-Aldrich.

### Virus strains, stock preparation, plaque assay, and *in vitro* infection

SARS-CoV-2-MUC-IMB-1, SARS-CoV-2-B.1.1.7, and SARS-CoV-2-GFP strains (Thi Nhu Thao et al, 2020) were produced by infecting VeroE6 cells cultured in DMEM medium (10% FCS, 100 μg/ml streptomycin, 100 IU/ml penicillin) for 2 days (MOI 0.01). Viral stock was harvested and spun twice (1,000 g for 10 min) before storage at −80°C. Titer of viral stock was determined by plaque assay. Confluent monolayers of VeroE6 cells were infected with serial five-fold dilutions of virus supernatants for 1 h at 37°C. The inoculum was removed and replaced with serum-free MEM (Gibco, Life Technologies) containing 0.5% carboxymethylcellulose (Sigma-Aldrich). Two days postinfection, cells were fixed for 20 min at room temperature with formaldehyde directly added to the medium to a final concentration of 5%. Fixed cells were washed extensively

with PBS before staining with $H_2O$ containing 1% crystal violet and 10% ethanol for 20 min. After rinsing with PBS, the number of plaques was counted and the virus titer was calculated. Viral RNA was isolated from harvested supernatant, and full-genome sequencing was performed in order to validate the absence of cell culture-mediated mutations (sequence available upon request).

Wild-type SARS-CoV-2 (Innsbruck 1.2), B.1.351, and B.1.617.2 isolates were grown on Vero-TMPRSS2/ACE2 cells and titrated via $TCID_{50}$ assay.

The work with primary SARS-CoV-2 isolates has been approved by the risk prevention service of the Technical University of Munich or the Medical University of Innsbruck—Institute of Virology and the Government of Upper Bavaria or the Austrian Labour Inspectorate in Tyrol. All work has been conducted according to BSL3 environment safety standards.

### Viral inhibitor assay and RT-qPCR analysis

A549-ACE2 cells were seeded into 96-well plates in a DMEM medium (10% FCS, 100 μg/ml streptomycin, 100 IU/ml penicillin) one day before infection. Six hours before infection, the medium was replaced by 100 μl of DMEM medium containing either the compounds of interest or the corresponding vehicle (DMSO) as a control. During postinfection treatments, 12 μl of DMEM medium containing either the compounds of interest 10-fold concentrated or the corresponding vehicle (DMSO) as a control was directly added 4 h postinfection to the infected wells containing 120 μl of the total medium. For the ADAM protease preactivation experiment, media containing PMA was exchanged after 30 min incubation prior to infection. Infection was performed by adding SARS-CoV-2-GFP (MOI 3) and plates were placed in the IncuCyte S3 Live-Cell Analysis System where real-time images of mock (phase channel) and infected (GFP and phase channel) cells were captured every 4 h for 48 h. Cell viability (mock) and virus growth (mock and infected) were assessed as the cell confluence per well (phase contrast area) and GFP area normalized on cell confluence per well (GFP area/phase contrast area) respectively using IncuCyte S3 Software (Essen Bioscience; version 2019B Rev2).

For comparative analysis of antiviral treatment activity against SARS-CoV-2 strains, A549-ACE2 cells were seeded in 12-well plates, as described above. Treatment was performed for 6 h with 1 ml of DMEM medium containing either the compounds of interest or the corresponding vehicle (DMSO) as a control and infected either with SARS-CoV-2-MUC-IMB-1 or SARS-CoV-2-B1.1.7 (MOI 1) for 24 h. Total cellular RNA was harvested and analyzed by RT-qPCR, as described above. 200 ng total RNA was used for reverse transcription with PrimeScript RT with a gDNA eraser (Takara). For relative transcript quantification, PowerUp SYBR Green (Applied Biosystems) was used.

| RPLP0_fwd | RefSeq ID: NM_001002, reference gene | GGATCTGCTGCATCTGCTTG |
|---|---|---|
| RPLP0_rev | RefSeq ID: NM_001002, reference gene | GCGACCTGGAAGTCCAACTA |
| 2019-nCoV_N-F | qPCR primers for SARS COV2 WHO & CRC recommended | TTA CAA ACA TTG GCC GCA AA |
| 2019-nCoV_N-R | qPCR primers for SARS COV2 WHO & CRC recommended | GCG CGA CAT TCC GAA GAA |

Experiments with Innsbruck and B.1.351 viruses were done as follows. One day prior to infection, $3 \times 10^4$ A549-ACE2 NTC cells/well were seeded in a 48-well plate. The following day, the medium was completely exchanged with 400 µl of fresh medium containing 10 µM of the indicated inhibitors or DMSO as control. Six hours afterwards, cells were infected either with wild-type SARS-CoV-2 or with B.1.351 variant at an MOI of 1. 24 h postinfection, medium was removed, cells were washed twice with PBS and lyzed with 50 µl DLR buffer (0.5% IGEPAL, 25 mM NaCl in 10 mM Tris-HCl Puffer + 1:100 Ribolock RNase inhibitor (TermoScientific, 40 U/µl)) for 15 min at room temperature. The suspension was centrifuged and the supernatant was used for RT-qPCR (iTaq™ Universal SYBR® Green One-Step Kit, BioRad) using one of the above primer sets.

For quantification of the secreted infectious particles by plaque assay, A549-ACE2 cells were seeded into 24-well plates in DMEM medium (10% FCS, 100 µg/ml streptomycin, 100 IU/ml penicillin) one day before infection. Six hours before infection, the medium was replaced with 500 µl of DMEM medium containing either the compound of interest or the vehicle as control. Infection was performed by adding SARS-CoV-2 (MOI 1) to the well and the infection was allowed to progress for 24 h. At that time, supernatants were harvested and frozen at −80°C until further use.

Confluent monolayers of VeroE6 cells were infected with serial five-fold dilutions of virus supernatants (from 1:100 to 1:7,812,500) for 1 h at 37°C. The inoculum was removed and replaced with serum-free MEM (Gibco, Life Technologies) containing 0.5% carboxymethylcellulose (Sigma-Aldrich). Two days postinfection, cells were fixed for 20 min at room temperature with formaldehyde directly added to the medium to a final concentration of 5%. Fixed cells were washed extensively with PBS before staining with $H_2O$ containing 1% crystal violet and 10% ethanol for 20 min. After rinsing with PBS, the number of plaques was counted and the virus titer was calculated.

## Generation of ADAM10 and ADAM17 CRISPR/Cas9 knock-out (CRISPR KO) constructs

Guide sequences for ADAM10 (guide 1: 5'-CGTCTAGATTTCCAT GCCCA-3' (exon 2), guide 2: 5'-GATACCTCTCATATTTACAC-3' (exon 3)), ADAM17 (guide 1: 5'-GGTCGCGGCGCCAGCACGAA-3', guide 2: 5'-GGTCTTTACCGAGTCTCTGG-3'; both exon 1), and a non-targeting control (NTC) sequence (5'-TCCGGAGCTTCTCCAG TCAA-3') were cloned into the vector lentiCRISPRv2 (Addgene, cat. no. 52961) according to the lentiCRISPRv2 and lentiGuide oligo cloning protocol (available on Addgene lentiCRISPRv2 website) with slight modifications as described in the following. Procedure for ADAM10 CRISPR KO constructs: 5 µg of the vector were digested with 1 µl BsmBI-v2 (NEB) in a total volume of 30 µl for 1 h at 55°C. Afterwards, 2 µl alkaline phosphatase (Roche) and 4 µl of the corresponding reaction buffer were added, and the reaction volume was adjusted to 40 µl by filling up with water, followed by 30 min incubation at 37°C. Both digestion and dephosphorylation were performed without DTT. Ligation of vector and oligos containing the guide sequence was carried out with T4 DNA ligase (Thermo) in a total reaction volume of 10 µl. Procedure for NTC and ADAM17 CRISPR KO constructs: 10 µl of the vector were digested with 6 µl BsmBI (NEB) in a final reaction volume of 60 µl. Oligo annealing was conducted in a total volume of 7 µl using 1 µl of 10× T4 DNA

ligase reaction buffer (Thermo) without the initial 37°C incubation step. Vector dephosphorylation and oligo phosphorylation were omitted. Ligation of vector and oligos was performed using T4 DNA ligase (Thermo) in a total reaction volume of 21 µl at 1–2 h of incubation.

All constructs were amplified using Stbl3 OneShot competent cells (Invitrogen) by adding 1–2 µl construct to 20–30 µl of the bacterial suspension. Competent cells were then incubated on ice for 30 min and transformed by heat shock at 42°C for 45 s. After cooling down on ice, bacteria were plated on ampicillin-containing agar plates. Next day, single colonies were picked for each construct and were further amplified in 2–6 ml of ampicillin-containing LB medium. Plasmids were extracted one day later, and the correct guide sequence was confirmed by Sanger sequencing using one of the following primers both located within the U6 promoter sequence: 5'-GAGGGCCTATTTCCCATGATTCCT-3', 5'-GACTATCA TATGCTTACCGT-3'.

To obtain ADAM10/17 double knock-out (KO) cells, the ADAM10 guide 1 sequence from the construct described above was cloned into the vector lentiCRISPRv2 hygro (Addgene, cat. no. 98291) by making use of NotI and EcoRV restriction sites as follows: 5 µg of either plasmid were supplemented with 2 µl restriction buffer 3.1 (NEB) and incubated with 0.3 µl NotI and 0.3 µl EcoRV-HF (both NEB) for 30 min at 37°C in a final reaction volume of 20 µl. Restriction fragments were separated on a 1% agarose gel. The fragment containing the ADAM10 guide sequence and the lenti-CRISPRv2 hygro counterpart including the hygromycin resistance cassette was excised and purified with the help of the NucleoSpin® Gel and PCR Clean-up kit (Macherey-Nagel) according to the manufacturer's instructions. 100 ng of the lentiCRISPRv2 hygro backbone were then ligated with 110 ng of the ADAM10 guide-containing fragment using 1 µl T4 DNA ligase (Thermo) in a total reaction volume of 10 µl. Incubation was performed at room temperature for roughly 3.5 h. 5 µl of the reaction was used for Stbl3 transformation and subsequent plasmid amplification as described above. The new construct was verified by sequencing as indicated above.

## Production of CRISPR/Cas9 KO lentiviral particles

HEK293T packaging cells were seeded on 10 cm culture dishes previously coated with poly-D-lysine (25 µg/ml of poly-D-lysine (Sigma) in PBS, incubation overnight at 37°C, dishes then washed twice with PBS) with $7 \times 10^6$ cells per dish and one dish for each CRISPR KO construct. Cells were incubated overnight at their usual growth conditions (37°C, 5% $CO_2$, DMEM (Thermo) supplemented with 10% fetal bovine serum (FBS, Thermo) and 100 µg/ml streptomycin, 100 IU/ml penicillin (Thermo)). 0.9 ml of Optimem (Thermo) were supplemented with 35.6 µl of Lipofectamine 2000 (Thermo), and liquids were mixed by inverting the tube. In a second tube, 0.9 ml of Optimem were mixed with 13.3 and 9 µg of the packaging plasmids pxPAX2 and pcDNA3.1-VSVG, respectively, and with 18 µg of the respective CRISPR KO construct. Both mixtures were pooled and then incubated at room temperature for 1 h. One hour prior to transfection, the HEK293T cell culture medium was changed to 9 ml of Optimem containing 10% FBS. The transfection mix was then added to the medium in a drop-wise manner, and cells were incubated overnight. On the next day, the medium was changed to 6 ml of DMEM supplemented with 2% FBS, 1%

penicillin/streptomycin, and 10 mM sodium butyrate. After overnight incubation, supernatants, which contained the lentiviral particles, were collected and centrifuged for 10 min at 1,000 $g$ in order to reduce the number of dead cells and debris. To remove cells and debris completely, supernatants were then passed through a 0.45 μm pore-size membrane filter (VWR) using a luer lock syringe. Supernatants were stored as aliquots of 1 ml at −80°C.

### Lentiviral transduction of ACE2-overexpressing A549 cells

For each CRISPR KO construct, a maximum of 850,000 ACE2-overexpressing A549 cells grown in T75 format were transferred into T25 flasks and incubated overnight at their usual growth conditions (37°C, 5% $CO_2$, DMEM with 10% FBS, and streptomycin/penicillin as described above). In an additional T25 flask, cells were seeded as a negative control for subsequent antibiotics selection. On the next day, the medium was replaced by 1 ml of the fresh normal growth medium, followed by adding 1 ml of the lentiviral supernatant of the respective CRISPR KO construct. Cells for selection control were supplied with 2 ml of the fresh medium instead. Polybrene was added to all cells at a final concentration of 5 μg/ml. After overnight incubation, cells were washed twice with PBS, and the medium was changed to 6 ml of normal growth medium supplemented with 1 μg/ml of puromycin or 500 μg/ml of hygromycin B, respectively, for selection. Antibiotics were withdrawn when the control cells were dead. For ADAM10/17 double KO generation, ADAM17 KO guide 1 cells were seeded in T25 flasks as described above and were co-infected with the lentiCRISPRv2 hygro ADAM10 KO construct. For all resulting cell lines, knock-out of the target gene was confirmed by Western blotting.

### MTT cell viability assay

To assess the toxicity of pharmacological compounds on A549-ACE2 cells, the CyQUANT MTT cell viability assay from Thermo Fisher (#V13154) was used according to the manufacturer's protocol. Briefly, $1 \times 10^4$ cells were seeded into a 96-well plate. After attachment, the cells were cultured in phenol-free DMEM (Thermo Fisher —21063029) containing 5% FBS and 0–25 μM BB94, DPC-333, apratastat, or DMSO for 48 h. Inhibitors were diluted so that each sample contained 0.25% DMSO in total. Two days after, cells were washed twice with phenol-free DMEM and incubated in 100 μl PBS containing 12 mM MTT for 4 h at 37°C. To dissolve the formazan, 100 μl of an SDS-HCL solution (1 g SDS dissolved in 0.01 M HCl) was added and incubated for 18 h at 37°C. Absorbance was measured at 570 nm.

### Western blot analysis

The A549-ACE2 cell lines were further characterized by Western blotting. Whole-cell lysates were obtained by exposing cells to ice-cold STET buffer containing a protease inhibitor cocktail (Roche, 1:500), 1 μM GI254023X, and 10 mM 1,10-phenanthroline. Lysates were cleared by centrifugation for 10 min at 11,000 $g$ at 4°C. Samples containing 10 μg of total protein were boiled for 5 min at 95°C in a 1× Laemmli buffer and subjected to an 8% SDS-PAGE. Antibodies used for detection are mentioned above.

### scRNA-seq analysis

Analyses of the scRNA-seq datasets including filtering, normalization, and clustering were conducted using Seurat 3.1 (Stuart et al, 2019). Human lung data from Han et al was downloaded from https://figshare.com/articles/HCL_DGE_Data/7235471 (Han et al, 2020a; Data ref: Han et al, 2020b) in the form of batch-corrected digital gene expression matrices and cell annotation csv files. Cell annotation included cell types, tissue of origin, and age. Gene expressions were log-normalized with a scale factor of 10,000 using the NormalizeData function. Next, data were scaled using the ScaleData function and the number of UMI and the percentage of mitochondrial gene content were regressed out as described by the authors (Han et al, 2020a). The first 40 principal components were considered for the UMAP (Uniform Manifold Approximation and Projection). Bronchoalveolar lavage fluid (BALF) data were acquired from GSE145926, and only the scRNA-seq data were used (Liao et al, 2020a; Data ref: Liao et al, 2020b). Data sets from 12 samples were quality-controlled, filtered, and integrated using the Seurat 3 standard integration workflow as described by the authors. Cell annotation of infection status, cell type, etc. was provided by the authors on their GitHub page (https://github.com/zhangzlab/covid_balf). For both data sets, DotPlot and DimPlot functions within Seurat were used for visualization purposes.

### Single-round infectious VSVΔG pseudotyped with SARS-CoV-2 spike protein

The single-round infectious VSVΔG with GFP as marker gene and pseudotyped with a C-terminally truncated SARS-CoV-2 spike protein (VSV-S) was generated as previously described (Riepler et al, 2020). As a negative control, a VSVΔG virus pseudotyped with the glycoprotein (GP) of the lymphocytic choriomeningitis virus (LCMV) was used (VSV-GP). One day prior to infection, $10^4$ cells in 100 μl complete medium were seeded per well of a 96-well plate. On the day of infection, 50 μl medium was removed from each well, and the inhibitors or DMSO were added in 50 μl to receive a final concentration of 10 μM (for E64d additional wells with a final concentration of 5 and 1 μM). After 1 h incubation at 37°C, cells were infected with 50 μl of the respective virus dilution aiming at a number of approximately 100 infected cells. The inhibitors were added a second time adjusting the concentration to the increased volume. For inhibition of viral infection with recombinant sACE2, infectious particles were incubated for 1 h prior to infection with half-logarithmic dilutions (9 steps, starting from 2.52 μg) of sACE2 (#RP-87703, Thermo Fisher) or BSA. 16 h after infection, infected, i.e., GFP-positive, cells were counted using an ImmunoSpot® S5 analyzer (C.T.L. Europe, Bonn, Germany).

### *In vitro* cleavage assay of spike protein

Recombinant spike protein (SIN-40589-V08B1-100, SinoBiologicals) was incubated with 1 μg of recombinant ADAM17 (Enzo Life Sciences) or ADAM10 (R&D) or TMPRSS2 (LSBio) for 24 h at 37°C at a 1:1 ratio. ADAM proteases were active in a 25 mM Tris buffer (2.5 μM $ZnCl_2$, 0.005% Brij-35 (w/v), pH 9.0). For TMPRSS2, the reactions were performed in a 50 mM Tris buffer (0.2% Triton X-100, 50 mM NaCl, pH 7.5). For inhibition of ADAMs and

TMPRSS2, proteases were incubated with BB94 (10 μM) or camostat mesylate (100 μM), respectively, for 5 min before the addition of spike protein. Samples were boiled for 5 min at 95°C in a 2× Laemmli buffer and subjected to an 8% SDS-PAGE. The cleavage fragments were detected using anti-HIS (Clone 3D5) antibody.

## Syncytium formation assay

HEK293T cells were co-transfected with pcDNA3.1(+)-eGFP and pCG1-SARS-CoV-2-S-HA (kind gift from S. Pöhlmann) using Lipofectamine 2000 according to the manufacturer's protocol. 18 h after transfection, cells were detached with PBS/EDTA (pH 7.5, 1 mM) and added on top of A549 cells stably expressing ACE2 at a 1:1 ratio ($4 \times 10^4$ cells/μ-Slide 8 Well, #80806, Ibidi). For inhibition of syncytium formation, 10 μM BB94 or DMSO was added at the time of co-cultivation. After 24 h, cells were fixed in 4% PFA for 15 min at room temperature and staining was performed in a blocking buffer (PBS, 2% NGS, 0.05% saponin). Cells were incubated with an anti-GFP antibody (rabbit, 1:500) for 45 min at room temperature, washed with PBS, and stained with secondary antibody (1:400, goat anti-rabbit Alexa Fluor 488, Abcam-ab150077), and Hoechst 33342 (1:2,000, Thermo Fisher-62249) for 30 min at room temperature. Images were recorded on a ZEISS LSM 900. Syncytia and nuclei were quantified by measuring the GFP and Hoechst fluorescence using Image J. To account for cell density differences, the area of syncytia relative to the nuclei intensity $A_{S,rel}$ was computed by $A_{S,rel} = A_{S,total} / I_{N,mean}$, where $A_{S,total}$ is the total area of syncytia and $I_{N,mean}$ is the mean intensity of nuclei. The relative proportion of nuclei within syncytia $P_N$ was calculated by $P_N = I_{N,ROI} * A_{N,ROI} / I_{N,mean}$, where $ROI$ is the region of interest, $I_{N,ROI}$ is the intensity of the nuclei within the $ROI$, and $A_{N,ROI}$ is the area of nuclei within the $ROI$, and $I_{N,mean}$ is the mean intensity of nuclei.

## Proximity ligation assay

A549-ACE2 cells were seeded into a 24-well plate containing a 15 mm coverslip glass. The next day, cells were infected with SARS-CoV-2-MUC-IMB strain at MOI 3 and placed on ice for 2 h. Cells were then moved to 37°C or left on ice for 15 min, washed with 1× PBS, and then fixed with 4% PFA. The Duolink proximity ligation assay (PLA) was carried out as described by the manufacturer (Sigma-Aldrich—DUO92007). For staining, the following antibodies were used: SARS Spike CoV-2 antibody [1A9] (GeneTex - GTX632604), ADAM17 antibody (Sigma-Aldrich—AB19027). All in a dilution of 1:100. Images were acquired on an LSM900 using a 63× oil immersion objective. Images were further processed using Image J. After background subtraction, nuclei and PLA foci were automatically quantified using the particle analyzer. For PLA quantification, four adjacent pixels were counted as foci. The number of foci per cell $F_{,rel.}$ was computed by $F_{,total} / N_{,total}$, where $F_{,total}$ is the total number of foci, and $N_{,total}$ is the total number of Nuclei present in an image.

## ELISA

For detection of soluble human TNFα, U937 cells were seeded at a density of $2 \times 10^5$ cells per 24-well and differentiated in RPMI (10% FBS, 100 μg/ml streptomycin, 100 IU/ml penicillin) containing 25 ng/ml PMA for 24 h. After that, cells were starved overnight and stimulated for 2 h with media containing 100 ng/ml LPS together with either DMSO or 10 μM of respective inhibitors. Supernatants were centrifuged for 5 min at 3,000 $g$ and then subjected to a V-PLEX Human TNFα ELISA from Meso Scale (#K151QWD) according to the manufacturer's protocol. For detection of soluble human APP-alpha (sAPPα) or soluble human APP-beta (sAPPβ), A549-ACE2 cells were seeded at a density of $2 \times 10^5$ cells per 24-well and incubated overnight in DMEM (10% FBS, 1% penicillin/streptomycin). Cells were cultured with fresh media supplemented with DMSO or 10 μM of respective ADAM protease inhibitors for 48 h. The BACE inhibitor C3 was used at 1 or 5 μM for 48 h. Supernatants were centrifuged for 5 min at 3,000 $g$ and analyzed using a human sAPPα ELISA Kit from IBL (#JP27734) or the sAPPβ kit from Mesoscale (K15120E-2) according to the manufacturer's protocol. For detection of soluble human ACE2 (sACE2), A549-ACE2 cells were seeded at $8 \times 10^5$ per 6-well and incubated overnight in DMEM (10% FBS, 1% penicillin/streptomycin). Cells were cultured with fresh media supplemented with DMSO, 10 μM BB94, or 5 μM DPC-333 for 48 h. PMA was spiked at 25 ng/ml to media containing DMSO 3 h before harvesting the supernatant. Supernatants were centrifuged for 5 min at 3,000 $g$ and analyzed using a human ACE2 DuoSet ELISA from R&D (DY933-05) according to the manufacturer's protocol.

## Submerged NHBE cultures

Primary human bronchial epithelial cells (NHBE, Lonza) from genetically independent donors were grown in monolayers in serum-free Bronchial Epithelial Cell Growth medium (BEGM, Lonza) as described elsewhere (Zissler *et al*, 2016). In order to avoid the influence of growth factors present in full growth medium (BEGM) on virus infection, cells were maintained in basal medium (BEBM) 24 h before infection. Six hours before infection, the medium was refreshed with a basal medium containing the respective compounds of interest. Infection with WT SARS-CoV-2-MUC-IMB-1 was performed by addition of 100,000 pfu per well. At the moment of harvest, cells were intensively washed with PBS and lysed for RT-qPCR analysis as described before.

## Gene expression analysis of NHBE cells

For whole-genome microarray analyses, mRNA from six independent NHBE donors in untreated condition was extracted using the RNeasy Micro kit (Qiagen, Hilden, Germany) according to the manufacturer's instructions. RNA integrity was confirmed using RNA 6000 Pico Chip Kit and Agilent 2100 Bioanalyzer (Agilent Technologies, Santa Clara, CA, USA) according to the manufacturer's instructions. Subsequently, microarray analysis was performed using single-color SurePrint G3 Human Gene Expression 8x60K Microarrays (Agilent Technologies, Waldbronn, Germany). The microarrays were performed and analyzed as described previously (Zissler *et al*, 2018). Data was acquired from GSE176405 (Jakwerth *et al*, 2022a; Data ref: Jakwerth *et al*, 2022b).

## Statistical analysis

All measurements were derived from distinct samples. Data processing, data visualization, and statistical analysis were performed in

Python 3.8 using pandas 1.1.4, matplotlib 3.3.4, seaborn 0.11, scipy 1.6.2, and statsmodels 0.12.2. For statistical testing of data with $N < 6$, Student's *t*-test was performed as recommended for extremely small sample sizes (de Winter, 2013). The Benjamini–Hochberg FDR method was used for multiple hypothesis correction (White *et al*, 2019) as implemented in statsmodels.stats. multitest.multipletests. One-way ANOVA and two-way ANOVA were computed as implemented in scipy.stats.f_oneway and statsmodels.stats.anova.anova_lm, respectively. Tukey's *post-hoc* multiple comparison test was used as implemented in statsmodels. stats.multicomp.pairwise_tukeyhsd. Adjusted *P*-values are reported in the figures. The 95% confidence interval (CI) was shown. Bar and line plots were created using seaborn (0.11.1).

## Data availability

This study includes no large data sets, and thus, no data are deposited in external repositories.

**Expanded View** for this article is available online.

## Acknowledgements
This work was funded by the Deutsche Forschungsgemeinschaft (DFG, German Research Foundation) under Germany's Excellence Strategy within the framework of the Munich Cluster for Systems Neurology (EXC 2145 SyNergy–ID 390857198), DFG research grants (PI 1084/3, PI 1084/4, PI 1084/5, TRR179/TP11, and TRR237/A07) and through the Federal Ministry of Research (BMBF) through projects CLINSPECT-M (FKZ161L0214C), JPND PMG-AD, ESCAPE (01KI20169A), and COVINET, by an ERC consolidator grant (ERC-CoG ProDAP, 817798) and the Bavarian State Ministry of Science and Arts (Bavarian Research Network FOR-COVID, ForCOVID/TP7). GJ was supported by a PhD scholarship from the Friedrich–Naumann–Stiftung für die Freiheit (FNF) with funds from the Bundesministerium für Bildung und Forschung (BMBF). Open Access funding enabled and organized by Projekt DEAL.

## Author contributions
**Georg Jocher:** Data curation; Formal analysis; Investigation; Visualization; Writing—review & editing. **Vincent Grass:** Investigation; Writing—review & editing. **Sarah K Tschirner:** Data curation; Investigation; Visualization; Writing—review & editing. **Lydia Riepler:** Investigation; Writing—review & editing. **Stephan Breimann:** Data curation; Formal analysis; Visualization. **Tuğberk Kaya:** Investigation. **Madlen Oelsner:** Investigation. **M Sabri Hamad:** Formal analysis. **Laura I Hofmann:** Visualization. **Carl P Blobel:** Writing—review & editing. **Carsten B Schmidt-Weber:** Supervision; Funding acquisition. **Ozgun Gokce:** Supervision; Funding acquisition; Writing—review & editing. **Constanze A Jakwerth:** Investigation; Writing—review & editing. **Jakob Trimpert:** Supervision; Funding acquisition; Investigation; Writing—review & editing. **Janine Kimpel:** Supervision; Funding acquisition; Writing—review & editing. **Andreas Pichlmair:** Supervision; Funding acquisition; Writing—review & editing. **Stefan F Lichtenthaler:** Conceptualization; Supervision; Funding acquisition; Writing—original draft; Project administration; Writing—review & editing.

In addition to the CRediT author contributions listed above, the contributions in detail are:
Conceptualization: SFL; Investigation: GJ, VG, SKT, LR, TK, CAJ, MO, JT; Formal analysis: SB, MSH; Writing—original draft: SFL; Writing—review & editing: GJ, VG, SKT, OG, CBS-W, JK, CPB, JT, AP, SFL; Visualization: SB, MSH, GJ, LIH; Supervision: OG, CBS-W, JK, JT, AP, SFL; Project administration: SFL; Funding acquisition: OG, CBS-W, JK, JT, AP, SFL.

## Disclosure and competing interests statement
The authors declare that they have no conflict of interest. Dr. Blobel holds a patent on a method of identifying agents for combination with inhibitors of iRhoms. Dr. Blobel and the Hospital for Special Surgery have identified iRhom2 inhibitors and have co-founded the start-up company SciRhom in Munich to commercialize these inhibitors.

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
