## [Review Process File · EMBO Reports]

ADAM10 and ADAM17 promote SARS-CoV-2 cell entry and spike protein-mediated lung cell fusion

Georg Jocher, Vincent Grass, Sarah Tschirner, Lydia Riepler, Stephan Breimann, Tugberk Kaya, Madlen Oelsner, Mohamed Hamad, Laura Hofmann, Carl Blobel, Carsten Schmidt-Weber, Oezguen Goekce, Constanze Jakwerth, Jakob Trimpert, Janine Kimpel, Andreas Pichlmair, and Stefan Lichtenthaler

DOI: [10.15252/embr.202154305](https://doi.org/10.15252/embr.202154305)

Corresponding author(s): *Stefan Lichtenthaler (stefan.lichtenthaler@dzne.de)* , *Andreas Pichlmair (andreas.pichlmair@tum.de)*

Review Timeline:

Submission Date:	10th Nov 21
Editorial Decision:	17th Dec 21
Revision Received:	11th Mar 22
Editorial Decision:	1st Apr 22
Revision Received:	7th Apr 22
Accepted:	12th Apr 22

Transaction Report:

Dear Mr. Jocher

Thank you for the submission of your manuscript to EMBO reports. So far, we have received two referee reports that are copied below.

Given that both referees are in fair agreement that you should be given a chance to revise the manuscript, I would like to ask you to begin revising your study along the lines suggested by the referees. Please note that this is a preliminary decision made in the interest of time, and that it is subject to change should the third referee offer very strong and convincing reasons for this. As soon as we will receive the final report on your manuscript, we will forward it to you as well.

Given the constructive comments we have thus far received, we would thus like to invite you to revise your manuscript with the understanding that the referee concerns (as detailed above and in their reports) must be fully addressed and their suggestions taken on board. Please address all referee concerns in a complete point-by-point response. Acceptance of the manuscript will depend on a positive outcome of a second round of review. It is EMBO reports policy to allow a single round of revision only and acceptance or rejection of the manuscript will therefore depend on the completeness of your responses included in the next, final version of the manuscript.

We realize that it is difficult to revise to a specific deadline. In the interest of protecting the conceptual advance provided by the work, we recommend a revision within 3 months (March 17, 2022). Please discuss the revision progress ahead of this time with the editor if you require more time to complete the revisions.

IMPORTANT NOTE:

We perform an initial quality control of all revised manuscripts before re-review. Your manuscript will FAIL this control and the handling will be DELAYED if the following APPLIES:

- 1) A data availability section providing access to data deposited in public databases is missing. If you have not deposited any data, please add a sentence to the data availability section that explains that.
- 2) Your manuscript contains statistics and error bars based on $n=2$. Please use scatter blots in these cases. No statistics should be calculated if $n=2$.

- Please consider point 10) regarding data references to cite Han et al., 2020 and Liao et al., 2020.
- Please identify the committee that approved the work with SARS-CoV-2 and add this and a statement whether the work confirmed to all safety standards in a BSL3 environment in the methods section.

- 1) a .docx formatted version of the manuscript text (including legends for main figures, EV figures and tables). Please make sure that the changes are highlighted to be clearly visible.
- 2) individual production quality figure files as .eps, .tif, .jpg (one file per figure). Please download our Figure Preparation Guidelines (figure preparation pdf) from our Author Guidelines pages <https://www.embopress.org/page/journal/14693178/authorguide> for more info on how to prepare your figures.
- 3) a .docx formatted letter INCLUDING the reviewers' reports and your detailed point-by-point responses to their comments. As part of the EMBO Press transparent editorial process, the point-by-point response is part of the Review Process File (RPF), which will be published alongside your paper.
- 4) a complete author checklist, which you can download from our author guidelines (). Please insert information in the checklist that is also reflected in the manuscript. The completed author checklist will also be part of the RPF.
- 5) Please note that all corresponding authors are required to supply an ORCID ID for their name upon submission of a revised manuscript (). Please find instructions on how to link your ORCID ID to your account in our manuscript tracking system in our Author guidelines
()

6) We replaced Supplementary Information with Expanded View (EV) Figures and Tables that are collapsible/expandable online. A maximum of 5 EV Figures can be typeset. EV Figures should be cited as 'Figure EV1, Figure EV2' etc... in the text and their respective legends should be included in the main text after the legends of regular figures.

7) Please note that a Data Availability section at the end of Materials and Methods is now mandatory. In case you have no data that requires deposition in a public database, please state so instead of refereeing to the database. See also < <https://www.embopress.org/page/journal/14693178/authorguide#dataavailability>>. Please note that the Data Availability Section is restricted to new primary data that are part of this study.

8) Figure legends and data quantification:

- the name of the statistical test used to generate error bars and P values,
- the number (n) of independent experiments (please specify technical or biological replicates) underlying each data point,
- the nature of the bars and error bars (s.d., s.e.m.)
- If the data are obtained from n {less than or equal to} 2, use scatter blots showing the individual data points.

9) We would also encourage you to include the source data for figure panels that show essential data. Numerical data should be provided as individual .xls or .csv files (including a tab describing the data). For blots or microscopy, uncropped images should be submitted (using a zip archive if multiple images need to be supplied for one panel). Additional information on source data and instruction on how to label the files are available .

10) Our journal encourages inclusion of *data citations in the reference list* to directly cite datasets that were re-used and obtained from public databases. Data citations in the article text are distinct from normal bibliographical citations and should directly link to the database records from which the data can be accessed. In the main text, data citations are formatted as follows: "Data ref: Smith et al, 2001" or "Data ref: NCBI Sequence Read Archive PRJNA342805, 2017". In the Reference list, data citations must be labeled with "[DATASET]". A data reference must provide the database name, accession number/identifiers and a resolvable link to the landing page from which the data can be accessed at the end of the reference. Further instructions are available at .

11) As part of the EMBO publication's Transparent Editorial Process, EMBO reports publishes online a Review Process File to accompany accepted manuscripts. This File will be published in conjunction with your paper and will include the referee reports, your point-by-point response and all pertinent correspondence relating to the manuscript.

Yours sincerely,

Referee #1:

The current study "The metalloproteases ADAM10 and ADAM17 control SARS-CoV-2 cell entry and spike protein-mediated lung cell fusion" by Jocher et al. identifies the matrix metalloproteases ADAM17 and ADAM10 as SARS-CoV-2 S-processing proteases. The authors show that MMP inhibition reduces SARS-CoV2 entry into target cells and negatively affects virus growth. While both, ADAM10 and ADAM17 are shown to proteolytically cleave S, the two proteases appear to fulfill different roles during SARS-CoV-2 infection and disease pathogenesis: It is indicated that ADAM17 promotes virus entry and therefore increases infectivity while ADAM10 is implicated in syncytia formation mediated by spike-ACE2 interaction.

The manuscript is clearly structured and well written. The presented findings are of interest for the field and contribute to our understanding of SARS-CoV-2 infection and COVID19 disease pathogenesis. While I am suggesting some additional experiments to improve the study I recommend the manuscript for publication.

Minor points

Fig 1a-g: The authors conclude that MMP inhibition affects entry of SARS-CoV-2 while their assay allows for multi-cycle growth of their reporter virus. I think this wording is not correct. In my view, only after including the pseudotyped VSV data (Fig1 h-i), the data allow for linking MMP inhibition to spike-mediated entry of SARS-CoV-2

Fig 2-3: When using SARS-CoV-2 clinical isolates, only qPCR data are shown. To fully assess the effect of MMP inhibition/depletion on virus infection growth curves showing virus titers would be desirable.

Fig 3F: the legend for the average gene expression is very small and difficult to read. Please enlarge the legend. Furthermore, besides ADAM17/10, also ACE2 expression is associated with disease severity in the presented dataset which should also be mentioned in the text to avoid overselling the contribution of the MMPs to disease severity/pathogenesis.

Major points

The authors hypothesize that MMP proteolytic processing of S occurs (in the case of A549-ACE2 cells) in the endosome or at the cell surface before endocytosis. This would be in line with earlier observations (and Fig 1d) that endocytosis is the major route of entry in A549 cells. Yet, the data do not show that the virus enters by endocytosis in their infection model. Nor do the authors show where the interaction between S and ADAM10/17 takes place (cell surface, endosome, or both). To investigate where the interaction takes place, proximity ligation assays and/or high resolution light or electron microscopy could be employed.

Given the surface expression of ADAM17, fusion at the plasma membrane is possible as well and could be an alternative entry route to endocytosis in the absence of cell surface TMPRSS2. It would be therefore beneficial to include co-treatments with cathepsin inhibitors together with the MMP inhibitors to determine any potential additive effect of targeting both groups of proteases in the A549 and primary cell model.

I understand that is this probably beyond the scope of the study but I believe that the study would benefit from including in vivo data. Using the hACE2 mouse model the (therapeutic) effect of the MMP inhibitors on SARS-CoV-2 infection could be studied as well as the syncytia formation in lung tissue. Alternatively, the therapeutic effects of the MMP inhibitors DPC-333 and apratastat in comparison or combination to TMPRSS2 and cathepsin inhibition could be investigated in the primary cell model with time-of-addition studies.

Referee #3:

Georg Jocher and colleagues demonstrate key roles for ADAM10 and ADAM17 in SARS-CoV-2 infection causing COVID19. These roles are independent of the protease TMPRSS2. ADAM17 controls viral entry into lung cells and ADAM10 regulates lung cell syncytia formation. This work is novel, important and will highly contribute to the fields of virus biology and proteolysis. After evaluating the manuscript, I found no major issues. In the spirit of improving the manuscript, I have 3 minor suggestions that the authors should consider addressing or changing.

Minor comments:

1- Figure 1d: TAPI-1 and E64d colors are challenging to distinguish. I would suggest changing one color. Would make it easier to distinguish for color-blind readers.

2- Figure 2b: the color between DMSO and DPC-333 are also hard to distinguish.

3- In Figure 5a and 5b, the recombinant spike protein was incubated with ADAM10, ADAM17 or TMPRSS2 at 1:1 ratio. Is this a physiological ratio as one patient would experience when infected with SARS-CoV-2? Often, a ratio of 1:100 (1 protease :100 substrates) tends to be considered more "physiologically relevant". Would a virus infection environment have more viral proteins as compared to host proteases? This could be commented in the discussion section.

We thank both Referees for their encouraging and constructive reviews. A point-by-point response to each of their comments is provided below.

Referee #1:

The current study "The metalloproteases ADAM10 and ADAM17 control SARS-CoV-2 cell entry and spike protein-mediated lung cell fusion" by Jocher et al. identifies the matrix metalloproteases ADAM17 and ADAM10 as SARS-CoV-2 S-processing proteases. The authors show that MMP inhibition reduces SARS-CoV2 entry into target cells and negatively affects virus growth. While both, ADAM10 and ADAM17 are shown to proteolytically cleave S, the two proteases appear to fulfill different roles during SARS-CoV-2 infection and disease pathogenesis: It is indicated that ADAM17 promotes virus entry and therefore increases infectivity while ADAM10 is implicated in syncytia formation mediated by spike-ACE2 interaction.

The manuscript is clearly structured and well written. The presented findings are of interest for the field and contribute to our understanding of SARS-CoV-2 infection and COVID19 disease pathogenesis. While I am suggesting some additional experiments to improve the study I recommend the manuscript for publication.

Minor points

Fig 1a-g: The authors conclude that MMP inhibition affects entry of SARS-CoV-2 while their assay allows for multi-cycle growth of their reporter virus. I think this wording is not correct. In my view, only after including the pseudotyped VSV data (Fig1 h-i), the data allow for linking MMP inhibition to spike-mediated entry of SARS-CoV-2

We fully agree with this comment. Thus, we rephrased the last sentence of this paragraph on page 5 and additionally replaced the word "entry" by "infection". The new sentence is now: "Taken together, SARS-CoV-2 infection of A549-ACE2 cells is facilitated by one or several metalloproteases".

We additionally changed the last sentence of the next paragraph (top of page 6): "We conclude that the metalloprotease-dependent viral entry into A549-ACE2 cells depends on the SARS-CoV-2 S protein and is observed for both SARS-CoV-2 and spike pseudotyped particles".

Fig 2-3: When using SARS-CoV-2 clinical isolates, only qPCR data are shown. To fully assess the effect of MMP inhibition/depletion on virus infection growth curves showing virus titers would be desirable.

As suggested by this reviewer, we determined virus titers by performing a plaque assay and included this new data into Figure EV1I. In agreement with the data obtained by qPCR, the plaque assay revealed that BB94 strongly inhibited the number of plaque-forming units compared to the control condition. A similar effect was observed for the lysosomal inhibitor E64d (also shown in new figure EV1I). We added the following sentence to the results on page 6: "The inhibitory effect of BB94 was also observed when using a plaque assay as a read-out to quantify the production of infectious particles and was similar to the effect of the lysosomal inhibitor E64d (Fig. EV1I)." Time-dependency of the infection, as measured with qPCR, is shown in Fig. EV1H and also demonstrates a clear reduction upon addition of BB94, apratastat or DPC-333.

Fig 3F: the legend for the average gene expression is very small and difficult to read. Please enlarge the legend. Furthermore, besides ADAM17/10, also ACE2 expression is associated with disease severity in the presented dataset which should also be mentioned in the text to avoid overselling the contribution of the MMPs to disease severity/pathogenesis.

As suggested, we increased the letter size for the legend and included the information on ACE2 both in Fig. 3F and in the results section on page 9, where we rephrased the original sentence. The new sentence is: "Most of the infected cells were found to express ADAM10 and/or ADAM17, while few cells expressed TMPRSS2 (Fig. 3E-G). Notably, expression of ADAM10 and ADAM17, and also of TMPRSS2 and ACE2 was higher in the severe COVID-19 patients compared to healthy controls (Fig. 3F)."

Major points

The authors hypothesize that MMP proteolytic processing of S occurs (in the case of A549-ACE2 cells) in the endosome or at the cell surface before endocytosis. This would be in line with earlier observations (and Fig 1d) that endocytosis is the major route of entry in A549 cells. Yet, the data do not show that the virus enters by endocytosis in their infection model. Nor do the authors show where the interaction between S and ADAM10/17 takes place (cell surface, endosome, or both). To investigate where the interaction takes place, proximity ligation assays and/or high resolution light or electron microscopy could be employed.

As suggested, we used a proximity ligation assay with an antibody to the spike protein and an antibody to ADAM17 and tested for interaction between S and ADAM17 in the viral infection assay using A549-ACE2 cells. After viral addition cells were kept for 2 h on ice (4°) to reduce endocytosis. Then, cells were shifted to 37°C for 15 min or kept on ice. PLA assay was performed after cells were fixed. We observed a PLA signal in the 4° condition and this signal was increased in the 37° condition. We conclude that the interaction between S and ADAM17 may start at the plasma membrane (4° condition) and is enhanced if endocytosis is allowed (37° condition). This new data is included as new Fig. 5C and is described in the results on page 10.

Additionally, we extensively tried to use electron microscopy but failed to convincingly detect sufficient numbers of viral particles to make statements on impaired engulfment of viral particles after hydroxamate treatment.

Given the surface expression of ADAM17, fusion at the plasma membrane is possible as well and could be an alternative entry route to endocytosis in the absence of cell surface TMPRSS2. It would be therefore beneficial to include co-treatments with cathepsin inhibitors together with the MMP inhibitors to determine any potential additive effect of targeting both groups of proteases in the A549 and primary cell model.

This is a very intriguing idea, which we experimentally tested: we performed the suggested experiment in both A549-ACE2 and the primary NHBE cells. We used the GFP-expressing SARS-CoV-2 virus and scored GFP accumulation as proxy for virus infection/replication. In both cell models the combined treatment with the cathepsin inhibitor E64d and the metalloprotease inhibitor BB94 reduced GFP fluorescence nearly completely and, thus, more strongly, than the individual treatments. Similar results were obtained for BB94 plus the TMPRSS2 inhibitor camostat in the NHBE cells (which express endogenous TMPRSS2). Thus, all three proteases (metalloproteases, TMPRSS2, cathepsins) contribute to SARS-CoV-2 infection in the primary cells, and their combined inhibition

completely blocks the infection. The new data is shown in Fig. 2D for the NHBE cells and in Fig. EV2C for the A549-ACE2 cells. In the results section on page 7 we added the following, new paragraph: "Similar results were obtained when infecting NHBE cells from different donors with the GFP-expressing SARS-CoV-2 virus and using GFP fluorescence intensity as a read-out (Fig. 2D). DPC-333 and apratastat treatment reduced GFP fluorescence intensity to a similar degree as treatment with BB94. In this dataset, more variability was observed for camostat and also for the lysosomal inhibitor E64d, which both reduced GFP signal by approximately 50%. Co-treatment with BB94 and either camostat or E64d or cotreatment with all three protease inhibitors (BB94, camostat and E64d) together efficiently reduced GFP expression (Fig. 2D), demonstrating the need for proteolytic cleavage for SARS-CoV-2 infection. Similarly, co-treatment of A549-ACE2 cells with BB94 and E64d blunted GFP-expression after infection with the SARS-CoV-2 reporter virus (Fig. EV2C)."

I understand that is this probably beyond the scope of the study but I believe that the study would benefit from including in vivo data. Using the hACE2 mouse model the (therapeutic) effect of the MMP inhibitors on SARS-CoV-2 infection could be studied as well as the syncytia formation in lung tissue. Alternatively, the therapeutic effects of the MMP inhibitors DPC-333 and apratastat in comparison or combination to TMPRSS2 and cathepsin inhibition could be investigated in the primary cell model with time-of-addition studies.

We fully agree that in vivo studies would be beneficial. However, such experiments would require intensive dosing optimization due to in-favorable pharmacokinetic properties of the hydroxamate-based metalloprotease inhibitors. Hydroxamates have, for instance, very short half-life in mouse models (e.g. Qian et al. 2007). In a pilot study we tested whether the half-life would be more favorable in hamsters, which are also used as animal models of COVID-19. We found that DPC-333 reached a maximum concentration in the low micromolar range within 30 min, but was rapidly cleared and barely detectable anymore 2 h after dosing. Due to legislative restrictions we could only apply the drug twice a day, which would only result in ADAM protease inhibition for about 2-4 hours per day. Formulation-based drug stabilization or continuous applications have been shown to aid the pharmacokinetic properties of certain drugs. However, we feel that this is beyond the scope of this manuscript, which describes ADAM10 and ADAM17 as alternative proteases to TMPRSS2. We consider hydroxamates as potentially highly valuable drugs to treat COVID-19 and will establish their therapeutic effects in appropriate infection models in the future.

As alternative to in vivo experiments, we followed the suggestion of the reviewer and established the therapeutic effects of the protease inhibitors in post-infection treatment regimes using ex vivo human primary cells and the human lung A549-ACE2 cells. We chose a setting, whereby the drugs were applied 4 h after infection (post-infection application) and compared the effects to the pre-infection application setting. We used the GFP-expressing SARS-CoV-2 virus. Post-infection application reduced infection to a similar extent as pre-infection application. However, the variation was somewhat higher, resulting in partly less efficient inhibition of viral infection. Overall, the inhibitors efficiently inhibited viral infection in both the pre- and the post-infection model. The new data is included in Fig. 2E, EV2D and described in the results section on page 7: "Post-infection drug addition resembles more closely a therapeutic situation in patients. To test whether the inhibitors are also active in post-infection scenarios we added them four hours after viral infection (post-infection) and monitored SARS-CoV-2-GFP growth. For both NHBE and A549-ACE2 cells the post-infection treatment with protease inhibitors had similar effects as compared to the pre-treatment drug addition, but resulted in greater variability among the donors," so that not all

treatments resulted in significant changes (Fig. 2E, Fig. EV2D). In conclusion, metalloprotease inhibitors that block ADAMs are effective in reducing viral load in primary human lung cells and are effective in post-exposure scenarios.”

As an additional experiment – although not specifically asked for by the reviewers – we included the currently predominant omicron variant of concern into the manuscript and demonstrate in Fig. 2A that BB94 is active to block infection of A549-ACE2 cells with this variant.

Referee #3:

Georg Jocher and colleagues demonstrate key roles for ADAM10 and ADAM17 in SARS-CoV-2 infection causing COVID19. These roles are independent of the protease TMPRSS2. ADAM17 controls viral entry into lung cells and ADAM10 regulates lung cell syncytia formation. This work is novel, important and will highly contribute to the fields of virus biology and proteolysis. After evaluating the manuscript, I found no major issues. In the spirit of improving the manuscript, I have 3 minor suggestions that the authors should consider addressing or changing.

Minor comments:

1- Figure 1d: TAPI-1 and E64d colors are challenging to distinguish. I would suggest changing one color. Would make it easier to distinguish for color-blind readers.

As suggested, we changed the color for TAPI-1 and checked the new colors with a software application for color-blind readers. We hope that the different lines are now better distinguishable.

2- Figure 2b: the color between DMSO and DPC-333 are also hard to distinguish.

As suggested, we changed modified the colors and checked the new colors with a software application for color-blind readers. We hope that the different bars are now better distinguishable.

3- In Figure 5a and 5b, the recombinant spike protein was incubated with ADAM10, ADAM17 or TMPRSS2 at 1:1 ratio. Is this a physiological ratio as one patient would experience when infected with SARS-CoV-2? Often, a ratio of 1:100 (1 protease :100 substrates) tends to be considered more "physiologically relevant". Would a virus infection environment have more viral proteins as compared to host proteases? This could be commented in the discussion section.

We agree with the reviewer that often a ratio of 1:100 is used, in particular for soluble proteases and their soluble substrates or when a soluble ADAM protease ectodomain is used with a short peptide-based substrate. However, the situation in cells is very different when considering that the full-length, membrane-bound protease (such as the ADAMs) and the full-length, membrane-bound substrate (such as the spike protein) come together. Here, the cleavage is not controlled by diffusion but rather by protein transport within the secretory pathway (if the cleavage happens in cis) or by the number of viruses that attach to the host cell surface, where cleavage may then happen in trans. Given that the total number of infected A549-ACE2 cells with our GFP-expressing SARS-CoV-2 virus was relatively low and

given that we used low MOIs for infections, we feel that a 1:1 ratio between protease and substrate is indeed approximating the conditions in our cell-based experiments. Yet, we have also tested the ADAM10 cleavage assay of the spike protein with a ratio of 1:20 (ADAM10:spike) and still detected cleavage, although to a lower extent as compared to the 1:1 ratio. Because we do not have exact data on the ratio of protease versus substrate in cells, we prefer to leave this discussion out of the manuscript.

Editorial office requirements for manuscript formatting:

1) A data availability section providing access to data deposited in public databases is missing. If you have not deposited any data, please add a sentence to the data availability section that explains that.

We have added a sentence on this topic into the manuscript.

2) Your manuscript contains statistics and error bars based on $n=2$. Please use scatter blots in these cases. No statistics should be calculated if $n=2$.

This point is not relevant to our manuscript, because n is at least 3 for all experiments.

- Please consider point 10) regarding data references to cite Han et al., 2020 and Liao et al., 2020.

We have included the link to the data references for both datasets.

- Please identify the committee that approved the work with SARS-CoV-2 and add this and a statement whether the work confirmed to all safety standards in a BSL3 environment in the methods section.

This sentence is included in the methods section.

1) a .docx formatted version of the manuscript text (including legends for main figures, EV figures and tables). Please make sure that the changes are highlighted to be clearly visible. We submit a track-changes version of the manuscript as well as a “clean” version, where all changes have been accepted. The changes include the new experiments as requested by the reviewers or formatting issues requested by the editorial office as well as some linguistic changes and corrections of typos.

2) individual production quality figure files as .eps, .tif, .jpg (one file per figure).

Please download our Figure Preparation Guidelines (figure preparation pdf) from our Author Guidelines pages

<https://www.embopress.org/page/journal/14693178/authorguide> for more info on how to prepare your figures.

Figures are uploaded in the required formats.

3) a .docx formatted letter INCLUDING the reviewers' reports and your detailed point-by-point responses to their comments. As part of the EMBO Press transparent editorial process,

the point-by-point response is part of the Review Process File (RPF), which will be published alongside your paper.

This is uploaded.

4) a complete author checklist, which you can download from our author guidelines (<https://www.embopress.org/page/journal/14693178/authorguide>). Please insert information in the checklist that is also reflected in the manuscript. The completed author checklist will also be part of the RPF.

This is uploaded.

5) Please note that all corresponding authors are required to supply an ORCID ID for their name upon submission of a revised manuscript (<https://orcid.org/>). Please find instructions on how to link your ORCID ID to your account in our manuscript tracking system in our Author guidelines

(<https://www.embopress.org/page/journal/14693178/authorguide#authorshippinguidelines>)

ORCID IDs have been added for Andreas Pichlmair and Stefan Lichtenthaler.

6) We replaced Supplementary Information with Expanded View (EV) Figures and Tables that are collapsible/expandable online. A maximum of 5 EV Figures can be typeset. EV Figures should be cited as 'Figure EV1, Figure EV2" etc... in the text and their respective legends should be included in the main text after the legends of regular figures.

Figures are now named accordingly.

<https://www.embopress.org/page/journal/14693178/authorguide#expandedview>

Not relevant for our manuscript.

This is relevant for our source data, which are uploaded accordingly.

7) Please note that a Data Availability section at the end of Materials and Methods is now mandatory. In case you have no data that requires deposition in a public database, please state so instead of refereeing to the database.

See also

< <https://www.embopress.org/page/journal/14693178/authorguide#dataavailability>).

Please note that the Data Availability Section is restricted to new primary data that are part of this study.

We have added a sentence on this topic into the manuscript.

8) Figure legends and data quantification:

- the name of the statistical test used to generate error bars and P values,
- the number (n) of independent experiments (please specify technical or biological replicates) underlying each data point,
- the nature of the bars and error bars (s.d., s.e.m.)
- If the data are obtained from n {less than or equal to} 2, use scatter blots showing the individual data points.

See also the guidelines for figure legend

preparation: <https://www.embopress.org/page/journal/14693178/authorguide#figureformat>

This has been updated in our manuscript.

9) We would also encourage you to include the source data for figure panels that show essential data. Numerical data should be provided as individual .xls or .csv files (including a tab describing the data). For blots or microscopy, uncropped images should be submitted (using a zip archive if multiple images need to be supplied for one panel). Additional information on source data and instruction on how to label the files are available <https://www.embopress.org/page/journal/14693178/authorguide#sourcedata>. Source data are uploaded.

10) Our journal encourages inclusion of *data citations in the reference list* to directly cite datasets that were re-used and obtained from public databases. Data citations in the article text are distinct from normal bibliographical citations and should directly link to the database records from which the data can be accessed. In the main text, data citations are formatted as follows: "Data ref: Smith et al, 2001" or "Data ref: NCBI Sequence Read Archive PRJNA342805, 2017". In the Reference list, data citations must be labeled with "[DATASET]". A data reference must provide the database name, accession number/identifiers and a resolvable link to the landing page from which the data can be accessed at the end of the reference. Further instructions are available at <https://www.embopress.org/page/journal/14693178/authorguide#referencesformat>. We have included the link to the data references for both datasets.

11) As part of the EMBO publication's Transparent Editorial Process, EMBO reports publishes online a Review Process File to accompany accepted manuscripts. This File will be published in conjunction with your paper and will include the referee reports, your point-by-point response and all pertinent correspondence relating to the manuscript. We agree with the RPF to be published.

Dear Prof. Lichtenthaler

Thank you for the submission of your revised manuscript to EMBO reports. We have now received the report from the referee who was asked to assess it and who recommends publication.

Browsing through the manuscript myself, I noticed a few editorial things that we need before we can proceed with the official acceptance of your study.

- Please reduce the number of keywords to 5.
- Please complete the funding info in the online submission system.
- Can you please split the source data into one file per figure? Maybe you can combine the figure source data with the p-values into one .xls file per figure?
- Acknowledgements, Author Contributions and the COI statement should be moved to after the Materials and Methods section.
- Please change the heading 'Declaration of interests' to 'Disclosure Statement and Competing Interests'. See also <https://www.embopress.org/competing-interests> for more information.
- Please add a heading 'Expanded View Figure Legends'.
- I attach to this email a related manuscript file with comments by our data editors. I also suggested some changes to title and abstract (present tense) and corrected the citations of datasets and preprints. Please address all comments and upload a revised file with tracked changes with your final manuscript submission.
- Finally, EMBO reports papers are accompanied online by A) a short (1-2 sentences) summary of the findings and their significance, B) 2-3 bullet points highlighting key results and C) a synopsis image that is 550x200-600 pixels large (width x height) in .png format. You can either show a model or key data in the synopsis image. Please note that the size is rather small and that text needs to be readable at the final size. Please send us this information along with the revised manuscript.

Yours sincerely,

Referee #1:

I thank the authors for addressing all points raised in the previous review round.
The manuscript is very interesting and I strongly recommend publication of the study.

The authors have addressed all minor editorial requests.

Prof. Stefan Lichtenthaler
DZNE and Technical University Munich
Neuroproteomics
Feodor-Lynen-Str. 17
Munich 81377
Germany

Dear Stefan,

Thank you for sending the missing source data files, which we have uploaded.

Now I am very pleased to accept your manuscript for publication in the next available issue of EMBO reports. Thank you for your contribution to our journal.

At the end of this email I include important information about how to proceed. Please ensure that you take the time to read the information and complete and return the necessary forms to allow us to publish your manuscript as quickly as possible.

As part of the EMBO publication's Transparent Editorial Process, EMBO reports publishes online a Review Process File to accompany accepted manuscripts. As you are aware, this File will be published in conjunction with your paper and will include the referee reports, your point-by-point response and all pertinent correspondence relating to the manuscript.

If you do NOT want this File to be published, please inform the editorial office within 2 days, if you have not done so already, otherwise the File will be published by default [contact: emboreports@embo.org]. If you do opt out, the Review Process File link will point to the following statement: "No Review Process File is available with this article, as the authors have chosen not to make the review process public in this case."

Should you be planning a Press Release on your article, please get in contact with emboreports@wiley.com as early as possible, in order to coordinate publication and release dates.

Please note that under the DEAL agreement of German scientific institutions with our publisher Wiley, your paper might be eligible for open access publication in a way that is free of charge for the authors. Please contact either the administration at your institution or our publishers at Wiley (emboreports@wiley.com) for further questions.
(see also <https://authorservices.wiley.com/author-resources/Journal-Authors/open-access/affiliation-policies-payments/institutional-funder-payments.html>)

Thank you again for your contribution to EMBO reports and congratulations on a successful publication. Please consider us again in the future for your most exciting work.

Kind regards,
Martina

THINGS TO DO NOW:

Please note that you will be contacted by Wiley Author Services to complete licensing and payment information. The required 'Page Charges Authorization Form' is available here: https://www.embopress.org/pb-assets/embo-site/er_apc.pdf

You will receive proofs by e-mail approximately 2-3 weeks after all relevant files have been sent to our Production Office; you should return your corrections within 2 days of receiving the proofs.

Please inform us if there is likely to be any difficulty in reaching you at the above address at that time. Failure to meet our deadlines may result in a delay of publication, or publication without your corrections.

All further communications concerning your paper should quote reference number EMBOR-2021-54305V3 and be addressed to

emboreports@wiley.com.

Should you be planning a Press Release on your article, please get in contact with emboreports@wiley.com as early as possible, in order to coordinate publication and release dates.